# Cost-effectiveness uncertainty may bias the decision of coal power transitions in China

Xizhe Yan [1], Dan Tong [1 ✉], Yixuan Zheng [2], Yang Liu[1], Shaoqing Chen[3,4], Xinying Qin[1], Chuchu Chen[2,5], Ruochong Xu[1], Jing Cheng [1], Qinren Shi[5], Dongsheng Zheng [1], Kebin He [5,6], Qiang Zhang [1] & Yu Lei [2 ✉]

A transition away from coal power always maintains a high level of complexity as there are several overlapping considerations such as technical feasibility, economic costs, and environmental and health impacts. Here, we explore the cost-effectiveness uncertainty brought by policy implementation disturbances of different coal power phaseout and new-built strategies (i.e., the disruption of phaseout priority) in China based on a developed unit-level uncertainty assessment framework. We reveal the opportunity and risk of coal transition decisions by employing preference analysis. We find that, the uncertainty of a policy implementation might lead to potential delays in yielding the initial positive annual net benefits. For example, a delay of six years might occur when implementing the prior phaseout practice. A certain level of risk remains in the implementation of the phaseout policy, as not all strategies can guarantee the cumulative positive net benefits from 2018–2060. Since the unit-level heterogeneities shape diverse orientation of the phaseout, the decision-making preferences would remarkably alter the selection of a coal power transition strategy. More strikingly, the cost-effectiveness uncertainty might lead to missed opportunities in identifying an optimal strategy. Our results highlight the importance of minimizing the policy implementation disturbance, which helps mitigate the risk of negative benefits and strengthen the practicality of phaseout decisions.

Despite the increasing demand for energy, there have been substantial reductions in air pollution emissions and $PM_{2.5}$-related health risks from the coal power sector in China that have resulted from a series of actions, including power fleet optimization, ultra-low emission standard implementation, and energy-saving renovation[1,2]. However, in 2017, the coal power fleet still contributes considerably to national emissions of carbon and air pollutants, namely, 35% of $CO_2$, 17% of $SO_2$, 19% of $NO_x$, and 8% of $PM_{2.5}$[3,4]. Under pressure to the increase of terminal electrification level and narrowed mitigation potential of end-of-pipe controls, the most feasible route to achieve the synergetic goal of carbon neutrality and clean air is to increase the pace of shrinking coal-dominated power systems[5–7]. Since the enactment of the 11th Five Year Plan (i.e., from 2006), China introduced a set of policies aiming at phasing out backward thermal power plants to address severe air

[1]Department of Earth System Science, Ministry of Education Key Laboratory for Earth System Modeling, Institute for Global Change Studies, Tsinghua University, Beijing, People's Republic of China. [2]State Environmental Protection Key Laboratory of Environmental Pollution and Greenhouse Gases Co-control, Chinese Academy of Environmental Planning, Beijing, People's Republic of China. [3]School of Environmental Science and Engineering, Sun Yat-sen University, Guangzhou, People's Republic of China. [4]Guangdong Provincial Key Laboratory of Environmental Pollution Control and Remediation Technology, Sun Yat-sen University, Guangzhou, People's Republic of China. [5]State Key Joint Laboratory of Environment Simulation and Pollution Control, School of Environment, Tsinghua University, Beijing, People's Republic of China. [6]Institute for Carbon Neutrality, Tsinghua University, Beijing, People's Republic of China. ✉e-mail: dantong@tsinghua.edu.cn; leiyu@caep.org.cn

pollution issues, leading to more than 119 GW of small, old, and inefficient coal-fired capacity being eliminated over this period[8]. The upcoming phaseout would pose a formidable barrier in a steady but evolutionary transition[9], as the majority of units were constructed around 2010 with similar lifespans[10]. In addition to prioritizing pollution control, the shifting policy preferences (e.g., health protection and climate mitigation) increase the complexity of phaseout decision[11,12]. It is difficult to determine the policy choices without capturing the patterns and uncertainties in prior practice. Thus, the future coal transition (i.e., the decision to phaseout and new-built power units) has been challenging in terms of finding probable and even promising cost-effective pathways. It is essential to fully consider the main factors of costs, climate impacts, and health burdens in shaping future turnover, which in turn necessitates a comprehensive assessment of the decision to phaseout and new construction.

The unit-level assessment of the coal power transition aligns with the current principles of refined governance in China. In response to various policy objectives, some global and regional studies have explored optimal coal power phaseout pathways based on the unit-level heterogeneities in health risks[13–15], technical attributes[16], and economic and environment impacts[17,18]. Despite holding considerable importance for public health and climate change mitigation, such single- and multi-path analyses tend to strictly follow the phaseout priority ranked by specific risks, which aims to reveal the maximized health co-benefit[13,14] or minimized asset stranding[10]. The practicality of applying these pathway analyses in designing the future transition roadmap may have limitations as the perfect management of unit phaseout. It means decision-making could be less reliable without the full evaluation of possible impact of policy implementation disturbance. In addition, new-built units are involved into stranded assets assessment of coal power transition[19], while location-based uncertainty of new-built units has not been unveiled, confounding the estimation of cost-effectiveness. Several studies have tried to manage the trade-offs among multiple policy objectives using multicriteria methods[20,21]; however, there is a limited understanding of the overall cost-effectiveness and uncertainty of various targeted phaseout strategies compared with historical strategies. As the design of coal power transition is at a critical juncture, there is an urgent need for an uncertainty assessment framework for various power fleet turnover trajectories that considers a full combination of objectives.

In this study, we developed an uncertainty assessment framework for multi-preference decision making at the unit level for the coal power transition in China (see Supplementary Fig. 1). Starting from the unit-level heterogeneities in stranded assets, carbon emissions, and health risks, we simulated the possible trends in different strategy-targeted pathways based on the prior phaseout practices from 2018–2060. We explored the cumulative risks and opportunities in those strategies while considering certain disturbance of phaseout policy implementation (i.e., the disruption of unit-level phaseout priority) and the uncertainty of new-built geolocations. In addition, we highlighted the potential impacts of cost-effectiveness uncertainty on the preference-based decision-making of coal power transition. In summary, unit-level information on carbon and air pollutant emissions, technical attributes (i.e., capacity, age, and coal consumption rate), and geolocations are directly obtained from the China coal-fired Power plant Emissions Database (CPED)[4]. Unit-level coal power-related deaths are isolated as potential phaseout health co-benefits, by using the Global Exposure Mortality Model (GEMM)[22] as the epidemiological exposure–response function and GEOS-Chem adjoint model[23] as sensitivity analysis tools. Under the harmonized provincial coal power demand projections[24] and the penetrations of Carbon Capture, Utilization and Storage (CCUS) in line with carbon peak and carbon neutrality goals[24,25], we defined five phaseout strategies (see Supplementary Table 1 in detail): the first allows power plants to operate for their historical expected lifetime (i.e., Historical strategy; assuming 40 years lifetime); the second remains consistent with the previous phaseout practice (i.e., business as usual, BAU strategy); the third gives priority to public health protection (i.e., Health strategy); the fourth targets climate change mitigation (i.e., Carbon strategy); and the final strategy targets economic loss avoidance (i.e., Age-to-Capacity strategy). Within strategies other than Historical, we assumed all existing coal capacity would be shut down by 2050 (an average lifetime of ~25.8 years). Future power fleet was simulated by incorporating an uncertainty simulation module, based on the Monte Carlo method, into a unit-by-unit coal power phaseout and new-built algorithm. We then examined how cost-effectiveness uncertainty might disrupt preference-based decision making of coal power transition by conducting a preference analysis using multiple criteria decision making (MCDM) methods. Finally, a series of sensitivity tests on the related factors (i.e., lower power demand, faster phaseout rate, higher health risk, and the CCUS priority) were conducted to reveal the potential variations of net benefits of different phaseout strategies, as well as to uncover the potential alternations in following preference-based decision makings (Supplementary Table 2).

## Results

### Unit-level heterogeneity

Figure 1 shows the geographical distribution of coal-fired power units and reveals heterogenous characteristics in 2018. Overall, China possesses a young coal power fleet with a large total installed capacity (Fig. 1a), accounting for 3.5 Gt of $CO_2$ emissions and 90,400 $PM_{2.5}$-related premature deaths in 2018. Due to the drastic variation among individual units, a small fraction of the generating capacity is disproportionately responsible for a large part of the climate threat (Fig. 1c) and health burden (Fig. 1d). For instance, approximately 13.7% of the total capacity is responsible for a quarter of $CO_2$ emissions. Similarly, 1.7% of the total capacity contributes to a quarter of air pollution-related deaths as there are relatively large disproportionalities between unit-level pollutant emissions and installed capacity (Supplementary Fig. 2). Across all categories of capacity, $CO_2$ emission intensity (defined as $CO_2$ emissions per capacity, highlighting the heterogeneity in $CO_2$ emissions) and the death intensity (defined as deaths per capacity, highlighting the heterogeneity in health risks; see "Methods" in detail) of small units (i.e., ≤100 MW) are 1.4 and 4.0 times larger than the national average level (Supplementary Fig. 3). These disparities can be attributed to the low penetration of advanced combustion technologies and end-of-pipe control measures in small units. By contrast, the distribution of the remaining unit-level assets exhibits a relatively even pattern, where 25% of the remaining assets are possessed by 20.3% of the capacity. This is because the majority of coal capacity (i.e., 71%) lies in units greater than 300 MW and younger than 15 years. These heterogeneous characteristics not only highlight the importance of a future customized phaseout of generating units, but also successfully reflect previous phaseout practice (Supplementary Fig. 4). Small, old, and inefficient units were prioritized for phaseout, leading to the retirement of 80% of units under 30 MW and more than 85% of units with coal consumption rates higher than 400 gce/kWh.

### Trends and uncertainties of different coal power transition strategies

Figure 2 shows the trends and uncertainty ranges in $CO_2$ emissions, deaths, and overall monetized benefits related to the coal power transition for each strategy. $CO_2$ emissions of coal power would peak at 4.1–4.3 Gt in 2030 and drop to 31–36 Mt (i.e., near zero, ~3% of total national emissions[25]) by 2060 (Fig. 2a), as the demand of coal power has drop to 8% of current level and the majority of coal power units are projected to complete CCUS retrofitting (~99%; see Supplementary Figs. 5 and 6). Such emissions from coal power would probably been offset by bioenergy with carbon capture and storage (BECCS) to

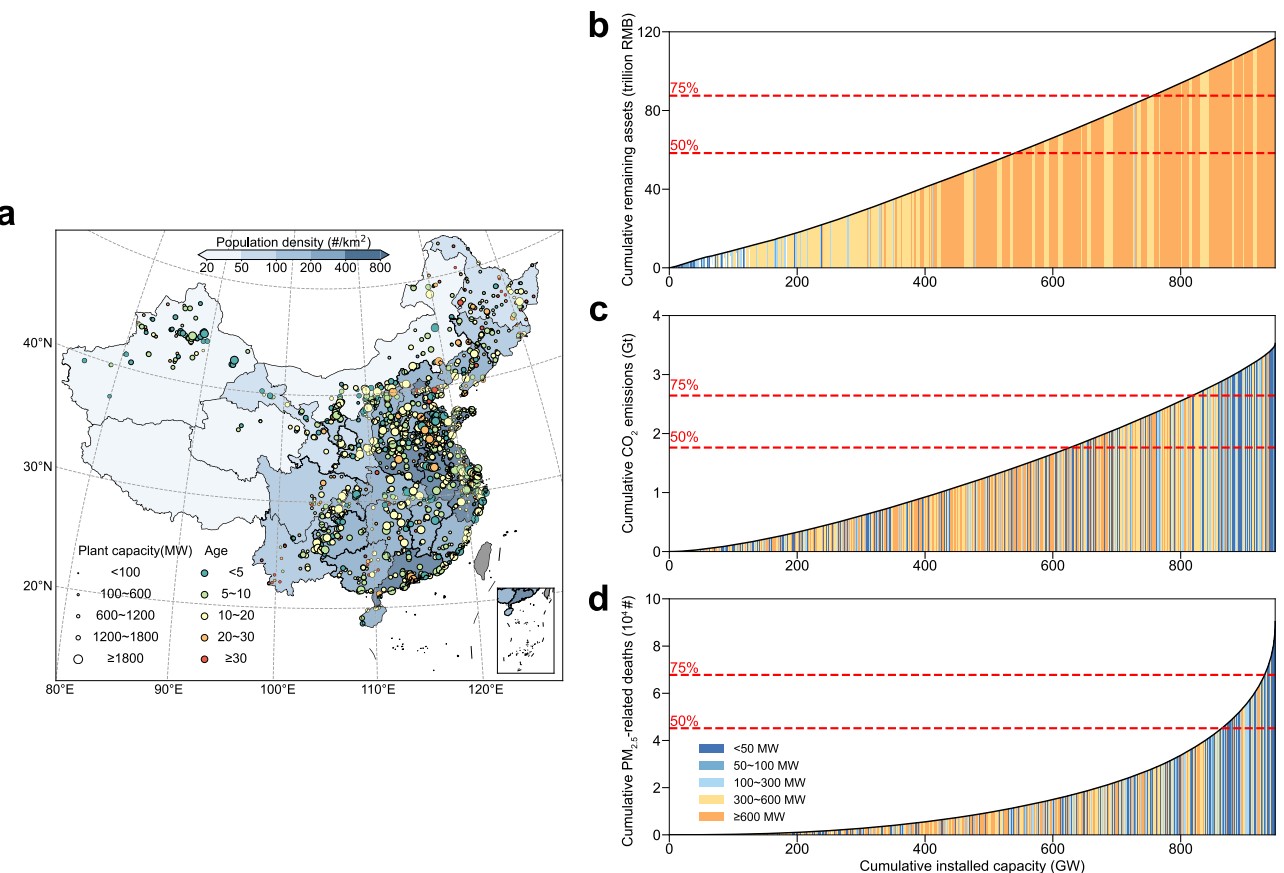

**Fig. 1 | Heterogenous characteristics of China's coal-fired power units. a** Geographical distribution, installed capacity and age of China's coal-fired generating units in 2018. **b–d** Ranking capacity by Age-to-Capacity ratio, carbon intensity, and death intensity reveals the drastic variation among individual generating units.

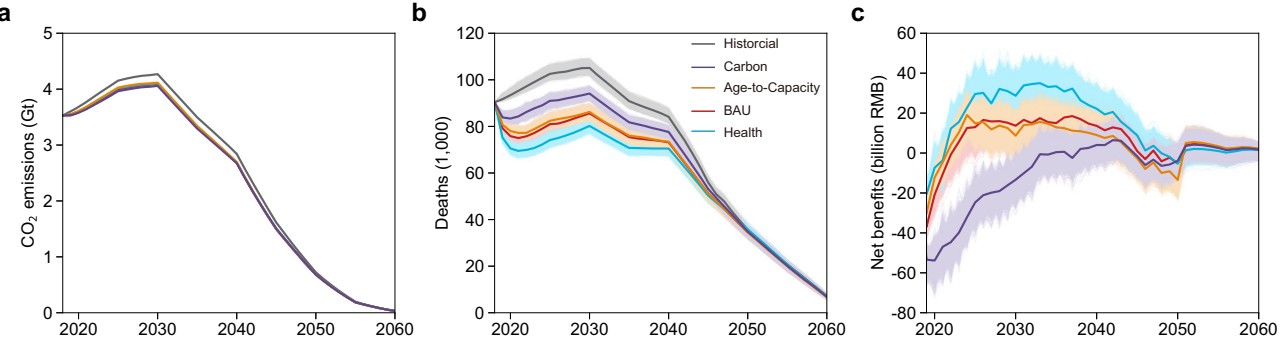

**Fig. 2 | Trends and possible ranges of CO$_2$ emissions, deaths, and benefits for each strategy. a** CO$_2$ emissions. **b** Coal-power-related premature deaths. **c** Monetized benefits compared to the historical phaseout pathway. The solid lines and the shaded area mean the median value and the uncertainty of each indicator under each strategy, respectively.

achieve carbon-negative in the power sector[26,27]. The differences in emission trends among strategies are relatively minor, with variations of less than 5% in 2030, due to the successful identification of part of low-efficient power units with higher CO$_2$ emission intensities in all strategies (Supplementary Fig. 2) and the same assumption of new capacity by ignoring the unit-level heterogeneity (i.e., 270 gce/kWh[28] in energy efficiency, see "Methods"). Nevertheless, Carbon strategy would hold in a most effective CO$_2$ emission reduction of 210 Mt compared to Historical strategy in 2030, which is around 1.4 times greater than that achieved through Age-to-Capacity strategy. It indicates that the strategic phaseout decision is still effective for deeper decarbonization. Distinct diversity exists in the death trends of different strategies, especially in the short and medium term

(i.e., 2019–2040; Fig. 2b). Compared to the Historical strategy, the annual avoided deaths driven by the Health strategy are 108% to 150% higher than those achieved by the Carbon strategy during 2019–2040 (Fig. 2b). This is due to its ability to capture the heterogeneous patterns of death intensity, indicating substantial health benefits when implementing health-targeted phaseout strategy.

All strategies likely bear unprecedented negative annual net benefits at the beginning stage (i.e., 2019–2021), and the timing for achieving a high-probability (>95%) positive annual net benefit varies when implementing the different strategies. Taking the Health strategy as an example, the annual net benefit is −20.6 (CI, −25.1 ~ −16.5) billion RMB in 2019 and soon reaches promising positive annual net benefits of 12.2 (CI, 4.9–19.3) billion RMB in 2022. Similar trends could be

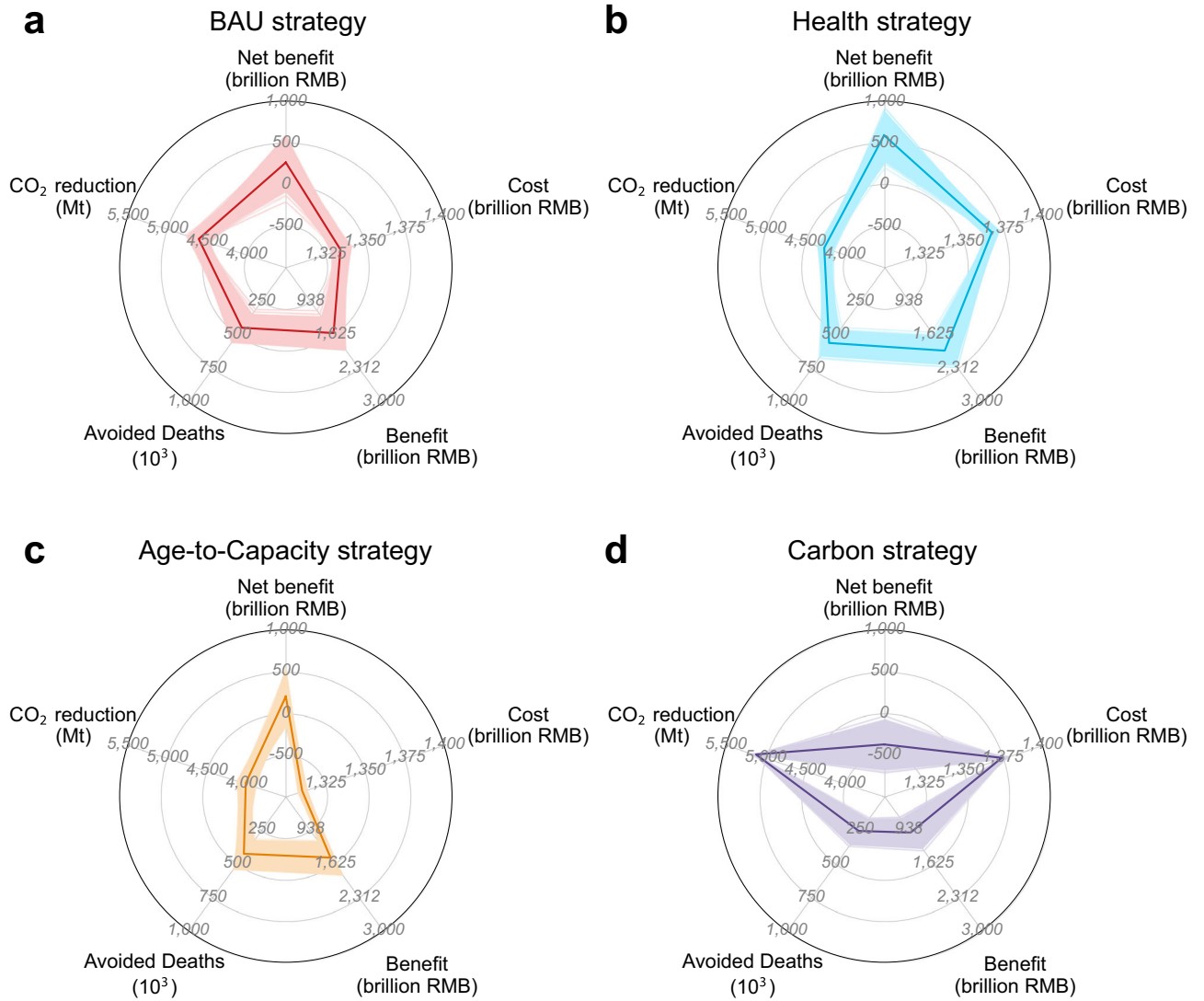

**Fig. 3 | Cumulative cost and benefits for each strategy. a–d** Cost represents the assets stranding of each strategy; and benefits represent the sum of monetized $CO_2$ emission reduction benefits and health co-benefits. The solid lines and the shaded area mean the median value and the uncertainty of each indicator under each strategy, respectively.

observed in the Age-to-Capacity and BAU strategies, but the latter could not reach a high-probability positive annual net benefit until 2024, revealing that targeted phaseout decisions would undergo an adaptation period in terms of net benefits during the early stage. This is because on one hand the implementation of the phaseout policy has to endure considerable economic losses in the near term, which would then be alleviated by the ageing process of the current young fleets. On the other hand, the economic losses would soon be exceeded by the gross benefits (except for the Carbon strategy), due to the structural changes (e.g., cleaner ones are kept) in power fleet driven by phaseout decisions. The possible trend in annual stranded assets of the BAU strategy lags behind that of the Age-to-Capacity strategy by approximately 3 years (Supplementary Fig. 7).

There is a potential delay in reaching the time in which the strategies start yielding positive annual net benefits as the disturbance of policy implementation. The attainment of initial positive net benefits from the Health and Age-to-Capacity strategies could be delayed by 3 years, while surprisingly, the BAU strategy might experience a 6-year delay (i.e., postpone from 2021 to 2027), as the potential gross benefits of the BAU strategy remain relatively comparable with the possible cost during 2021–2027. It is noted that, by 2030, 30–55% of the total

benefit uncertainty is contributed by the disturbance of phaseout policy implementation (Supplementary Fig. 8). While, as the majority of existing units were phased out around 2050, the increasingly dominant role of new-built capacity in supply future coal power generation within all targeted strategies (from ~65% in 2030 to ~85% in 2040), especially under the anticipate of reductions in utilization rate of both existing and new capacity (i.e., more new-built capacity is needed to supply the same generation demand; Supplementary Text 1; Supplementary Fig. 9), reveals that new-built capacity does holds importance in shaping net benefit and its associated uncertainty. Therefore, both phaseout and new construction decisions should be cautiously planned due to its vital role in quickly and continuously achieving positive net benefits.

## Cumulative cost-benefits and uncertainties

Figure 3 illustrates the cumulative costs and benefits from 2018–2060, whose differences among the strategies would be further amplified. In terms of public health protection, for example, the greatest health co-benefit is achieved through the Health strategy, cumulatively avoiding over 563,000 (CI, 515,900–611,000) premature deaths, while the Carbon strategy performs worst, avoiding 259,200 (CI,

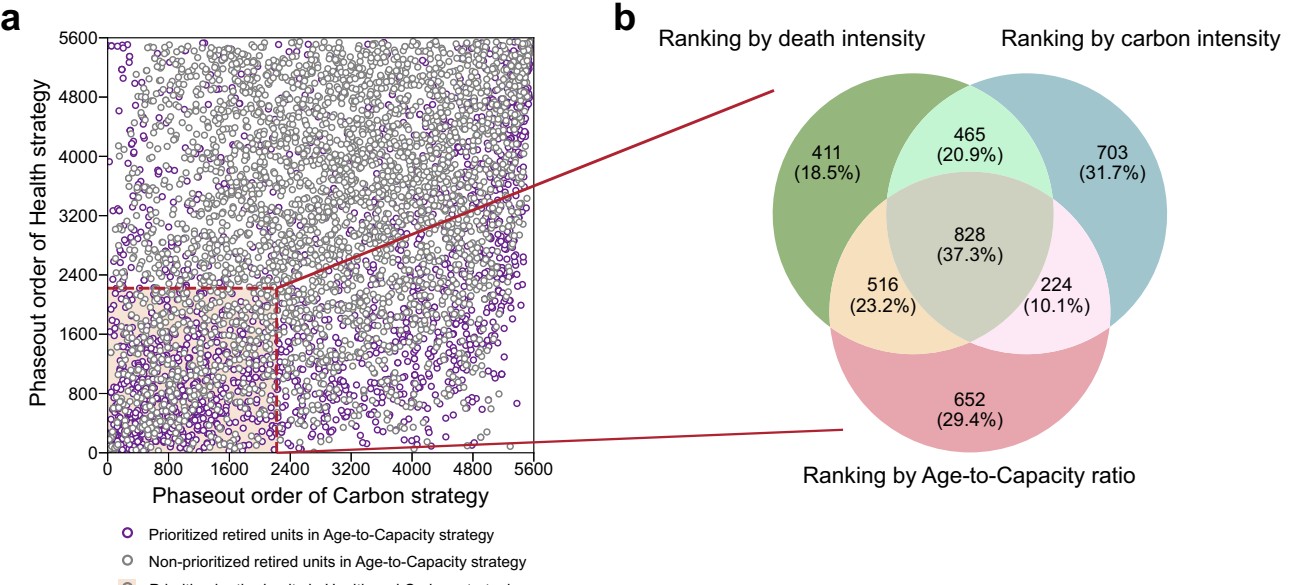

**Fig. 4 | Source of the disparities in net benefits among strategies. a** The phaseout priorities of each unit driven by Health, Carbon, and Age-to-Capacity strategies. The dots in pink shaded area represent prioritized retired units (ranking < 40th percentile) in both Health and Carbon strategies, and the purple dots represent prioritized retired units in Age-to-Capacity strategy. **b** Venn diagrams showing the numbers of shared and unique prioritized retired units among three phaseout strategies.

213,100–305,000) cumulative premature deaths. In addressing $CO_2$ emission mitigation, Carbon strategy has the potential to achieve a cumulative reduction exceeding 5 Gt, when the energy efficiency of new capacity meets the standard of 270 gce/kWh, which is around 1.2 times greater than that of Age-to-Capacity strategy. Taking the advancements in combustion technology driven by phaseout into consideration, the cumulative decarbonization efforts are further amplified among all targeted strategies (Supplementary Text 2; Supplementary Fig. 10), if Historical strategy remains current average energy efficiency (i.e., 307.6 gce/kWh[29]). Especially in the Carbon strategy, $CO_2$ emissions are cumulatively reduced by over 7 Gt during 2018 to 2060. It indicates that both strategic phaseout and energy efficiency improvements would make a difference in decarbonization.

A large disparity in the cumulative monetized net benefits arises across different strategies. For example, the Health strategy yields the highest cumulative monetized net benefits of 587.9 (CI, 435.7–739.3) billion RMB among all the strategies, while the Carbon strategy has the lowest (−378.1 billion RMB; CI, −527.7 - −230.6). The high level of value of a statistical life (VSL) amplifies the potential for a health-focused phaseout strategy to achieve maximal overall net benefits, while the current low carbon price remains an obstacle for a climate-focused strategy to attain positive net benefits (see Supplementary Tables 3 and 4; "Methods"). Carbon strategy is likely to achieve positive cumulative net benefits, which would be comparable to the cumulative net benefits of other strategies if the European carbon price (75.5 USD/t $CO_2$) is implemented (Supplementary Fig. 11).

A certain level of risk still remains in the phaseout policy implementation. When conducting the BAU or Age-to-Capacity strategy, there is a certain probability of achieving cumulative negative net benefits (Fig. 3a, c). It means that targeted phaseout strategies do not guarantee a 100% positive outcome brought by policy implementation disturbance. When it comes to a faster phaseout rate to close the prior practice (i.e., 30-year lifetime for Historical strategy and assuming an entire phaseout of existing units by 2040 for all targeted strategies), the risk of negative outcomes would be amplified by rapidly diminishing the benefit gap between Historical and other strategies (Supplementary Text 3; Supplementary Fig. 12). In addition to designing a proper phaseout strategy, measures such as strategizing the reasonable introduction of new-built units, maintaining the current level of annual utilization hours for advanced units, and extending the lifespan of advanced units by prioritizing the CCUS installation would be useful to mitigate the risk of cumulative negative outcomes and maximize net benefits (Supplementary Texts 4 and 5; Supplementary Figs. 13–15).

The phaseout priorities of coal power units conflict across distinct strategies (Fig. 4). Limited correlations ($R^2 < 0.2$) in the ranking orders are observed among the Health, Carbon, and Age-to-Capacity strategies. For example, 20% to 30% of the prioritized retired units (defined as units with ranking orders below the 40th percentile) are unique to their respective strategies (Fig. 4b), which indicates that different strategies would restructure the coal power fleet in entirely divergent directions, underscoring the importance of choosing and insisting appropriate strategies to meet the long-term strategic needs. In comparison, the phaseout priorities of the Age-to-Capacity and BAU strategies are much closer (Supplementary Fig. 16), leading to similar cumulative monetized net benefits. However, 37.3% (i.e., 828 units) of prioritized retired units are shared among the Health, Carbon, and Age-to-Capacity strategies. Those shared units exhibit poor performance in all policy objectives and should be promptly decommissioned as low-hanging fruits[16].

## The bias of strategy selection based on decision preferences

The unit-level heterogeneous characteristics shape the orientation of diverse decision making and guide the design of phaseout strategies tailored to varying preferences (Fig. 5). Specifically, the benefits of climate mitigation (weighed by $\alpha$), stranded asset avoidance ($\gamma$), and health co-benefits ($\beta$) could all serve as potential direct or indirect benefits from the coal power phaseout. That is, $CO_2$ emission reduction would represent the direct benefits of the climate-targeted coal power phaseout policy, while health co-benefits and economic loss would constitute its indirect benefits. A dimensionless indicator (named the normalized net benefit) obtained by normalizing and weighed summing the above three factors can assist decision makers in strategy assessment based on their time-varying preferences

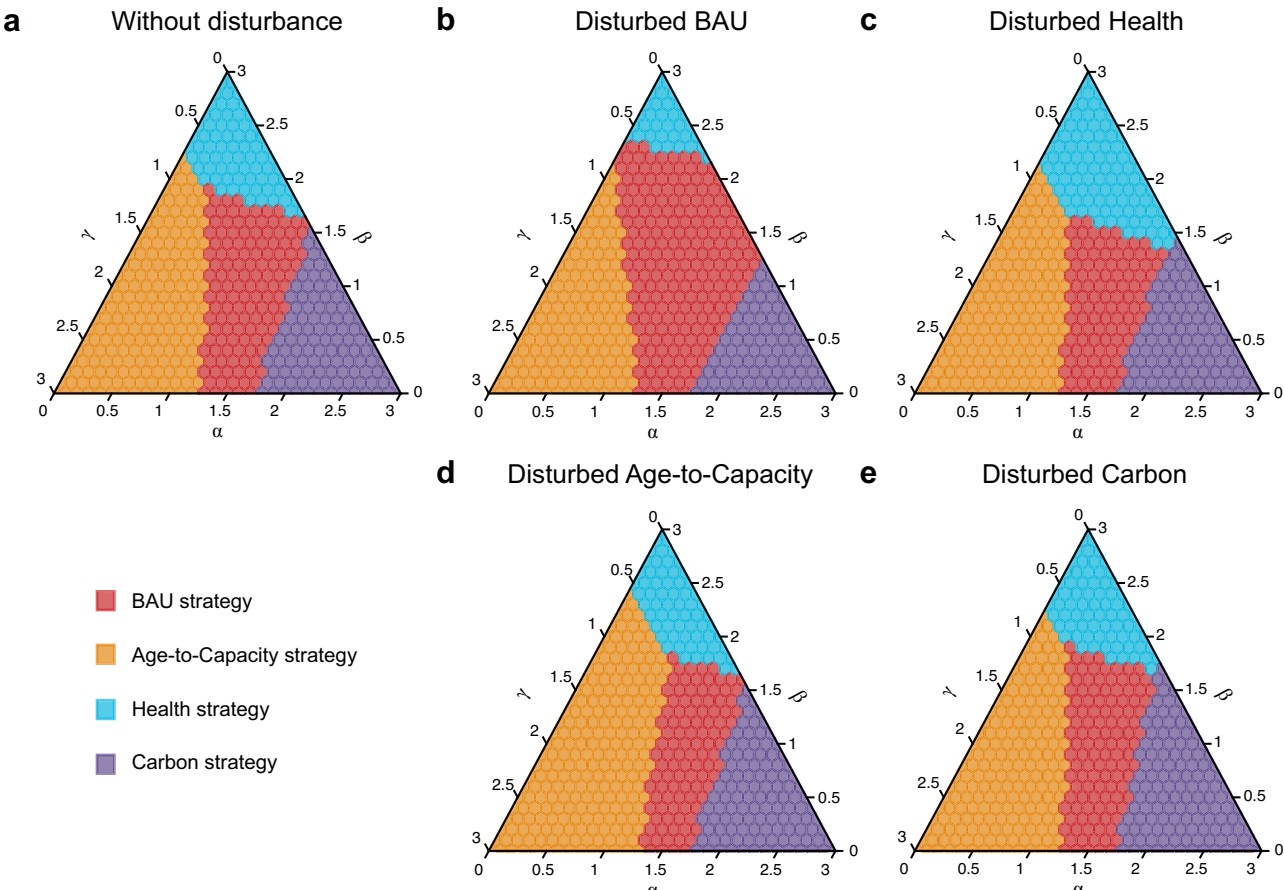

**Fig. 5 | Preference analysis for coal power phaseout strategy design. a** Coal power phaseout strategy selection according to the maximum normalized net benefits. $\alpha$, $\beta$, and $\gamma$ represent the preference weighting factor of climate change mitigation, public health protection, and assets stranding avoiding, respectively. The higher value of the specific preference weighting factor, the more emphasis is placed on that preference. **b**–**e** Disturbed phaseout strategy decisions, which are disturbed by the pathway with 95th percentile of monetized net benefits under BAU (business as usual, **b**), Health (**c**), Age-to-Capacity (**d**), and Carbon (**e**) strategies, respectively.

(Supplementary Fig. 17). Taking the Health strategy as an example, the normalized net benefit of the Health strategy is particularly sensitive to $\gamma$ (preference of cost saving). Health strategy could lead to substantial losses if costs are prioritized over the health co-benefits; conversely, there is the potential to yield considerable gains because the Health strategy has relatively higher health co-benefits and implementation costs compared to other strategies, which indicates that Health strategy should be chosen only when health protection becomes the primary objective. Regarding the Age-to-Capacity strategy, the normalized net benefits are relatively insensitive to changes in $\gamma$, remaining positive among almost all preferences due to its lowest economic loss. It reveals that the Age-to-Capacity strategy is relatively conservative, although it may not be the optimal strategy under certain preferences.

Phaseout decisions should be tailored to policy preferences (Fig. 5a). It is advisable to execute a phaseout policy following the Age-to-Capacity strategy and prioritize the decommissioning of old or small generating units with fewer remaining assets under decision making that emphasizes cost savings (e.g., $\gamma > 2$). In addition, in the context of an ageing population and the healthy China initiative, safeguarding public health might also become a primary preference in coal power phaseout (e.g., $\beta > 2$ and $\gamma < 0.5$). The Health strategy would be the optimal choice for promptly decommissioning super-polluting units located in densely populated areas. In some situations (e.g., $\alpha$, $\beta$ and $\gamma \sim 1$), the current strategy (i.e., BAU) has its advantages. This is because its principle of prioritizing the phaseout of small and inefficient units allows it to balance emission reduction and economic loss avoidance, indicating that the prior phaseout strategy might be feasible and effective in the future transition.

The uncertainty of policy implementation would might lead to a decision deviating from the optimal strategy to address specific policy objectives, and bias the strategy selection in the preference-based decision making (Fig. 5b–e). Given a specific preference (e.g., $\alpha$, $\gamma = 0.6$ and $\beta = 1.8$), there is a certain possibility for the BAU strategy to miss the opportunity for maximum net benefits; instead, the Health or Age-to-Capacity strategy might be better. And uncertainty would still have an undeniable impact on the preference-based decision making at the local scale (the case study in Inner Mongolia; Supplementary Text 6; Supplementary Fig. 18). Therefore, the phaseout decision should be carefully designed according to the decision preferences of policy makers while minimizing the disturbance in policy implementation to optimize overall benefits and alleviate the risk of uncertainty.

## Discussion

In this research, our uncertainty assessment framework for the coal power transition decision in China simulates future possible turnover at the unit level in support of carbon peak and carbon neutrality goals by 2060, estimates the annual cost effectiveness of each possible turnover, and reveals the potential impact of policy implementation uncertainty on coal transition decision making. Phaseout strategies are tailored based on the unit-level heterogeneity compatible with

different policy preferences. Within this framework, our evaluation implies that not all the phaseout strategies can guarantee positive cumulative net benefits. Rather, the extent to which cumulative net benefits of phaseout policy could ultimately be realized might hinge on the following key factors: one is the tailored phaseout strategy corresponding to decision preference; one is the limited disturbance of policy implementation; one is the strategic design of new power unit construction; and one is the time-varying operating factors for existing units (e.g., shifts in phaseout rate, and utilization hours).

A series of sensitivity tests are carried out to explore potential variations related to operating factors (Supplementary Table 2), which has been presented in the Supplementary Information in detail. It is noted that the tests of CCUS priority (i.e., allocating 50% of the CCUS demand to new-built units and assign the remaining demand to existing units) and stable annual utilization hours (i.e., remaining the current level annual utilization hours) show that the net benefits of targeted phaseout would increase within all strategies (see Supplementary Texts 4 and 5). It's because well-performing existing units with lower phaseout priorities have a chance to retain their roles in the mid- or long-term. Especially in the test of CCUS priority, more than 250 GW of existing units could be involved into CCUS retrofitting, and keep operating beyond their assumed lifetimes (15-year protection as investment recovery period of CCUS retrofitting[30,31]). Those units would continuously expand the disparities in cost-effectiveness among strategies tailored to different preferences and ultimately make strategies other than BAU becomes more competitive across preferences (Supplementary Fig. 19).

In alignment with the historical patterns of phaseout decision, BAU might remain the most appropriate and feasible phaseout strategy in future coal power transition, for its advantages in balancing emission mitigation and economy. However, considering the changing preferences and above-mentioned time-varying external factors, phaseout decision is still an open issue. Especially when it dives to the local level, decision makers should tailor phaseout plans according to the specific characteristics of coal power fleet. For example, Age-to-Capacity strategy might be more suitable for Inner Mongolia than BAU strategy as its young and huge coal fleet (Supplementary Text 6; Supplementary Fig. 20).

Our study has some limitations and uncertainties that require deeper exploration in future endeavors. First, various coal power generation scenarios (i.e., reflecting the degrees of clean energy development) may introduce uncertainties due to the insufficient temporal resolution of integrated assessment models (IAMs) and the overlooked feedback of changing cost effectiveness when implementing different phaseout strategies. Our test of a lower coal power demand (82% of original demand in 2030) indicates that the relative gaps of benefits and costs among strategies has not changed noticeably as the same assumption for phaseout and new-built mechanism (Supplementary Text 7; Supplementary Figs. 21 and 22). Second, baselined emission dataset (i.e., CPED) contains a small number of blank spots and missing information (Supplementary Table 5) and it can comprehensively capture the key features of coal power fleet in China after appropriate imputation (Supplementary Texts 8 and 9). Third, future retrofits of end-of-pipe control measures in coal power are disregarded in our study when focusing on phaseout policy assessment, as the meet of ultra-low emission standard for most coal-fired power units. The further stringency of pollution controls would have limited mitigation potential and may result in considerable costs and $CO_2$ emissions. Fourth, health risk assessment based on GEMM and GEOS-Chem adjoint model may arise certain uncertainties for their incomplete understanding of mortality risk, chemical and physical processes, future meteorological conditions, and structural evolution in the population, leading to a potential underestimation of the health co-benefit[32] (Supplementary Text 10; Supplementary Fig. 23). Finally, we ignored the uncertainty distribution of parameters (e.g., emission

factor) to emphasize the cost-effectiveness uncertainty caused by phaseout decisions.

Regardless, by modeling the possible turnover of coal power fleet at the unit level, our results add important nuance to policy-relevant discussions of the potential impact of cost effectiveness uncertainty in the decision of coal power transition. In support of the synergetic governance of the power sector, our data-driven dynamic uncertainty assessment framework for coal power transition may not only reveal the complexity of phaseout policy but also provide decision makers with a more comprehensive understanding of the risk and potential outcomes of preference-based decisions. Recognizing the dynamic nature of phaseout decision affected by variable factors such as socioeconomic changes, technology innovation, and implemented policies, we will incorporate the most up-to-date unit level information into our modeling framework and keep tracking the cost-effectiveness uncertainty of preference-based coal transition decision in a 5-year step (in line with the Five-year Plans of China). This, in turn, enhances the robustness, adaptivity, and applicability of cautiously designed phaseout strategies, ensuring that they remain promisingly cost-effective in response to different policy preferences.

## Methods

### Unit-level emission database and premature deaths isolation
We extract emissions data, technical attributes, and locations from the unit-level database, then estimate the premature mortality attributed to long-term $PM_{2.5}$ exposure and isolate the unit-level coal power-related health burden as potential phaseout health co-benefit.

We employ unit-based information (i.e., locations, unit capacity, start year of operation, coal consumption rate, and $CO_2$ and pollutants emission) from CPED. Constructed based on unit-level information sourced from the Ministry of Ecology and Environment (MEE, unpublished data), CPED is a high-resolution emission inventory that provides year-by-year emissions from coal-fired power plants since 1990 in China developed by Tsinghua University[4,33]. More details about CPED can be found in Supplementary Text 8 and a detailed description about the preprocessing of baseline emission inventory has been added in the Supplementary Text 9.

The assessment of premature mortality attributable to $PM_{2.5}$ exposure necessitates an understanding of the relationship between chronic exposure and response (C-R). Burnett et al.[34]. developed integrated concentration-response functions (IER) for the Global Burden of Diseases Study (GBD), based on cohort studies of ambient air pollution, first- and second-hand tobacco smoking, and household indoor air pollution. Although IER are widely applied to calculate $PM_{2.5}$-related mortality in research community[35,36], non-ambient $PM_{2.5}$-mortality associations in IER functions and additional causes of death other than the five (i.e., ischemic heart disease, stroke, chronic obstructive pulmonary disease, lung cancer, and lower respiratory infections) considered by the GBD may bias mortality estimation, especially in highly polluting region like China. To resolve those uncertainties, premature mortality attributable to ambient $PM_{2.5}$ exposure is estimated by using the GEMM in our study, which was developed by Burnett et al.[22] in 2018 across almost the entire global $PM_{2.5}$ exposure range, especially the polluted areas like China. GEMM is applied to estimate the hazard ratios between long-term $PM_{2.5}$ exposure and all the non-accidental deaths due to non-communicable diseases (NCDs) and lower respiratory infections (LRIs). The relative risk ($RR$) of NCD + LRI on $PM_{2.5}$ concentration ($C$) is calculated as:

$$RR(C) = e^{\frac{\theta \times \ln(1 + \frac{z}{\alpha})}{1 + \exp(\frac{\mu - z}{\upsilon^2})}}, z = \max(0, C - 2.4) \tag{1}$$

where $z$ represents the maximum of 0 and ($C$−2.4). No risk is assumed below the counterfactual concentration of 2.4 μg/m$^3$. $e$ represents Euler's number, and $\theta$, $\alpha$, $\mu$, and $\upsilon$ are parameters that determine the

shape of PM$_{2.5}$-mortality function. In GEMM framework, the *RR* of NCD + LRI is estimated by age for adults aged from 25 to 85, with 5-year intervals. The attributable fraction (*AF*) of mortality is converted by relative risk as:

$$AF(C) = \left(\frac{RR(C) - 1}{RR(C)}\right) \qquad (2)$$

Following Geng et al.[37], premature mortality attributable to PM$_{2.5}$ exposure (*M*) for a population subgroup *s* (divided by age) in grid *j* as:

$$M_{s,j}(C_j) = P_j \times PS_s \times B_s \times AF_s(C_j) \qquad (3)$$

where $P_j$ represents the total population in grid *j*, $PS_s$ are the national percentage fraction of a population subgroup s to total population, $B_s$ is national cause-specific baseline mortality incidence rate of NCD + LRI for population subgroup *s* and $AF_s(C_j)$ is the attributable fraction of NCD + LRI at PM$_{2.5}$ exposure level $C_j$ for population subgroups. 2018-specific spatial distribution of population ($P_j$) is from LandScan global population database (https://landscan.ornl.gov/)[38] and ground-level PM$_{2.5}$ concentration ($C_j$) is derived from Tracking Air Pollution in China dataset (TAP; http://tapdata.org.cn/)[39]. Both $PS_s$ and $B_s$ are derived from Global Burden of Disease Study 2019 (https://ghdx.healthdata.org/gbd-2019)[40].

Unit-level health burden is isolated by GEOS-Chem adjoint model, on this basis of PM$_{2.5}$ related mortality. Widely used in source apportionment, the adjoint of GEOS-Chem is able to calculate the response of PM$_{2.5}$-related premature deaths to the change in emission of major PM$_{2.5}$ precursors (i.e., SO$_2$, NO$_x$, NH$_3$), carbonaceous particles (i.e., OC, BC) and primary dust. In this work, we use GEOS-Chem adjoint v35i model to perform nested China (15°S–55°N, 70°E–150°E) simulation driven by the Multi-resolution Emission Inventory of China (MEIC: http://www.meicmodel.org/)[3] at a horizontal resolution of 0.25° lat × 0.3125° lon. The dynamic boundary condition is derived from global 2° lat × 2.5° lon simulation driven by Community Emissions Data System (CEDS) inventory[41]. GEOS-FP data from Global Modeling and Assimilation Office (GMAO) is used as the meteorological input of both nested and global simulations. To reduce computation cost, we conducted 4 months simulations (January, April, July, and October of 2018) for each season and their results are averaged to represent the annual level adjoint sensitivity result in 2018. GEMM PM$_{2.5}$-related premature mortality, as Eqs. (1–3), is defined as the cost function in this adjoint model. The output of adjoint simulation provides the partial derivatives ($\frac{\partial M}{\partial E_{j,k}}$) of total premature deaths in China (*M*) to emission in grid *j* ($E_{j,k}$; *k* represents species). The precursor species considered in this study are SO$_2$, NO$_x$, NH$_3$, BC, OC and primary PM$_{2.5}$. Following Zhao et al.[36], we combine the sensitivity and gridded emission to obtain the semi-normalized sensitivity (*SS*):

$$SS_{j,k} = \frac{\partial M}{\partial E_{j,k}} \times E_{j,k} \qquad (4)$$

*SS* is further normalized to calculate the contribution of grid-specified emission to premature deaths:

$$P_{j,k} = \frac{SS_{j,k}}{\sum_j \sum_k SS_{j,k}} \times 100\% \qquad (5)$$

According to previous studies, the normalization process minimizes the nonlinear effect among emission change, air quality and mortality[42]. Integrating unit-level emission information taken from CPED[4], we attributes premature deaths into each coal-fired power unit,

following Eq. (6):

$$M_i = M^* \sum_k (P_{j,k}^* \beta_{i,k}) \qquad (6)$$

where $M_i$ represents premature deaths caused by pollutants emission from units *i*, $\beta_{i,k}$ represents the emission ratios of species *k* caused by unit *i* in grid *j*.

In pervious study, emission intensity (tonnes per MW) is used to depict the relationship between generating capacity and environment impact from each coal-fired units and further identify the super-polluting units[18,43]. Inspired by definition of emission intensity, we defined death intensity (deaths per MW) as unit-level PM$_{2.5}$-related premature deaths ($M_i$) per installed capacity ($C_i$) to highlight the notable heterogeneity in health risks among units[7], following Eq. (7):

$$Death\ intensity_i = \frac{M_i}{C_i} \qquad (7)$$

## Strategy design and uncertainty framework for coal power transition

The structure and characteristics of the future coal power fleet would reshape driven by phaseout decision. In this study, five coal power phaseout strategies are designed considering the prior phaseout practice and various heterogeneities. For the Historical strategy, the 40-year lifetime is set for all units, according to the historical lifespan of global power plants and reflecting the economic consideration of operating costs, replacement costs, and revenues. For the BAU strategy, phaseout policy is implemented as the previous phaseout practice following the phaseout priority obtained from Cox model. For the Carbon strategy, units with large value of carbon intensity are prioritized for phaseout, aiming at deep decarbonization. For the Health strategy, units with large value of death intensity are prioritized for phaseout to maximize the health co-benefits. For the Age-to-Capacity strategy, units with large ratio of age to capacity size are prioritized for phaseout, in order to mitigate the risk of stranding assets as possible.

Since 11$^{th}$ Five Year Plan, small, old, and inefficient units were prioritized for phaseout in most case (Supplementary Fig. 4). In order to simulate the phaseout strategy as prior practice, we conduct survival analysis based on proportional hazards regression (also known as Cox regression)[33]. Taking the key technical attributes (i.e., age, installed capacity and coal consumption rate) into consideration, the survival outcomes of all operating and retired in-fleet coal power units in CPED are brought into this model as training data. The function of Cox regression in our study is shown as:

$$[b, logl, H, stats] = coxphfit([Var_{cap}, Var_{corat}], age, 'censoring', censor) \qquad (8)$$

where *b* represents coefficient estimates, *logl* represents log likelihood value, *H* represents estimated baseline cumulative hazard; *stats* represents coefficient statistics. coxphfit represents cox proportional hazards regression, censoring indicator for censoring $Var_{cap}$ and $Var_{corat}$ represent installed capacity and coal consumption rate of coal-fired power units, *age* represents lifetimes for the retired units and operated years for the in-fleet units. *censor* is the indicator for censoring by using 1 for the in-fleet units and 0 for the retired units.

The concordance index (C-index) is not only a metric which is used to assess the goodness of fit of a Cox model similar to R-squared, but also an indicator which represents the ability of the model to correctly provide a reliable ranking of the survival probability based on the individual risk scores. The C-index value is 0.68 in this model, and the function analog effect is good. Based on the construction of this Cox function, we predict the survival curve of in-fleet operating units and determine their phaseout order by their median age of retirements. This approach helps to depict the future phaseout trajectory

following the previous decisions orientation (BAU strategy), and determine a 32% margin of disturbance in the phaseout priorities when policy implementation.

Constrained by provincial projection of future coal power generation under the carbon peak and carbon neutrality goals (Supplementary Fig. 5), a possible power fleet projection model that couples phaseout algorithm with Monte Carlo framework is constructed to simulate the lifespan and operating status of each unit on an annual basis. To reflect disturbance of phaseout policy implementation, the model first divides coal-fired power units into ten groups according to the phaseout priority of specified strategy; and then randomly selects 32% (corresponding to C-Index estimated by Cox model) of the units in each group and disturbs their phaseout priorities. The sensitivity test for the setting of disturbance groups is tested in Supplementary Text 2. For each strategy, the possible future power fleet turnover is projected according to a set of runs ($n = 10{,}000$) in the Monte Carlo framework with uncertain disruptions of the original phaseout priorities.

Then, within each province in China, for a given year, the phaseout algorithm retires a certain coal power capacity following each possible phaseout priority of specific strategy, and calculates the deliverable power generation within the existing power fleet after implementing phaseout policy, which is determined by the installed capacity and capacity factors of in-fleet units. Considering the structural changes and the flexibility transformation of coal power, the current existing units in China would be completely phaseout in 2050[44], while the unit-level capacity factors is configured to decay by a rate of 2.5% annually. If the existing units is unable to meet the provincial power demand, new-built units with higher efficiency and advanced emission control device (see Supplementary Table 6) would put into commission in order to fill the generation gap; else, the capacity factors of current existing units would be adjusted to satisfy the balance of generation and load. The site selection for new-built units is randomly selected from the locations of retired units. Note that the future coal power fleet is driven by the coal power demand projection, and specific phaseout decision (i.e., shifting coal power fleet to other low-carbon energy sources) is out of our discussion.

Based on the unit-level projection of generation and capacity factors, we further model the emission and health burden of current existing units and new-built units. For current existing units, the unabated emission factors of $SO_2$, $NO_x$, $PM_{2.5}$, BC, and $CO_2$ are assumed to remain constant. The emission and death caused by current existing units are directly calculated by the changing rate of capacity factors. The $NO_x$ emission factors for new power units are obtained from the CPED for corresponding boiler size (i.e., 600 MW) and combustion technology (i.e., ultra-supercritical). The provincial average emission factors from the CPED are adopted for the $SO_2$, $PM_{2.5}$, BC, and $CO_2$ emission factors of new units. Then, we calculate $SO_2$, $NO_x$, $PM_{2.5}$, BC, and $CO_2$ emissions for new-built units by using the following equation:

$$E_{s,i} = G_i \times P \times \frac{H_0}{H_k} \times EF_{s,k} \times (1 - \eta_s) \times 10^{-6} \quad (9)$$

where $s$, $k$, $i$ represent emission species, province and new-built unit, respectively. $E$ represents unit-level emissions (kg), $G$ represents specific power generation for each unit (kWh); $P$ is the coal consumption rate (gce/kWh); $H$ represents the provincial average heating value of coal used in power generation (kJ/g); $H_0$ is the heating value of standard coal (29.27 kJ/gce), and the ratio of $H_0$ to $H$ converts the coal equivalent (gce) to the physical quantity of coal (gram). $EF$ represents the provincial unabated emission factors (g/kg); and $\eta$ represents the removal efficiency meeting ultra-low emission standards (details in Supplementary Table 6). The estimations of health burden for new-built units are consistent with aforementioned

procedure (details in Unit-level emission database and premature deaths isolation section).

A method framework of CCUS retrofit in Chinese coal-fired power units is shown in Supplementary Fig. 24. We first combine the future harmonized coal power demand projection and the CCUS retrofitting ratio (see Supplementary Fig. 6) explored by Cheng et al., 2021[25] to obtain possible supply curve of provincial CCUS retrofit in coal power. Most provinces are slated to initiate large-scale commercialization of CCUS around 2030, with the majority of coal capacity completing retrofitting by 2060. Similar to the phaseout algorithm, a CCUS retrofitting algorithm is further designed to retrofit a certain coal power capacity within each province in China for a given year, according to each possible phaseout priority of specific strategy and unit-level generation projection. To minimize the risk of stranding assets resulting from CCUS retrofitting, we assume that the installation requirements for CCUS would be prioritized for new-built capacity. If the new-built units could not meet the supply curve of Coal-CCUS demand, old units with less phaseout priority are prioritized for CCUS retrofitting to fill the gap. A sensitive test of the allocation of CCUS installation requirements among new-built and existing capacities is further discussed in Supplementary Text 5.

Although CCUS could substantially reduce $CO_2$ emissions, the additional electricity demand as CCUS systems in the future may have a non-negligible impact on fuel consumption and coal-related health risk[45,46]. In this study, coal power units with CCUS installation are adopted to achieve a $CO_2$ capture rate of 90% with an additional 15% energy consumption. Furthermore, it is reported that the unit capital cost of CCUS retrofitting would vary from 3300 RMB/kW to 8500 RMB/kW[30,47]. A potential life extension of 15 years is allocated to units with CCUS installation, aligning with retrofitting investment recovery periods that takes the high cost of capital cost, operating and maintenance cost into consideration[30,31]. The potential impact of life extension is then feedbacked to adjust the Coal-CCUS demand gap.

## Benefit assessment and preference analysis

The cost effectiveness of phaseout policy in our study is represented by the monetized net benefit ($Benefit_{net}$). We calculate the monetized net benefit for all possible turnover, which is consists of economic benefits of decarbonization ($Benefit_{decarbonization}$), health co-benefits ($Benefit_{health}$), and assets stranding ($Cost_{assets\ stranding}$), using following equation:

$$Benefit_{net,i,y} = Benefit_{decarbonization,i,y} + Benefit_{health,i,y} \\ - Cost_{assets\ stranding,i,y} \quad (10)$$

where $i$ and $y$ represent the specific phaseout turnover and year, respectively. We estimate annual economic benefits of decarbonization by the $CO_2$ emission reduction of turnover $i$ compared to Historical strategy as:

$$Benefit_{decarbonization,i,y} = \left( E_{Historical,y} - E_{i,y} \right) \times CP \quad (11)$$

Carbon price ($CP$, in units of RMB per $tCO_2$) is assumed as 50 RMB per $tCO_2$, according to the average trading prices of emission right of National Carbon Emission Trading System in 2021[48,49]. A sensitivity test of implementing European carbon price is performed to assess the impacts if carbon price is higher in the future (Supplementary Text 2). The monetized health co-benefits of each pathway are estimated by the number of deaths avoided and $VSL$ as:

$$Benefit_{health,i,y} = \sum_p \left( \left( Death_{Historical,y,p} - Death_{i,y,p} \right) \times VSL_p \right) \quad (12)$$

where $p$ represents the specific province in China. VSLs in all provinces are obtained from previous literature (Supplementary Table 4)[50,51], which are based on willingness to pay method in China and adjusted by provincial GDP per capita values. The annual assets stranding of each turnover is calculated by Eq. (13)[52]:

$$Cost_{assets\,stranding,i,y} = \sum_r OCC \times C_{i,y,r} \times \frac{(LT - Age_{i,y,r})}{LT} \qquad (13)$$

where $r$ represents the specific coal power unit retired in year $y$ in turnover $i$; $OCC$ represents overnight capital cost (RMB/MW), which is compiled from previous literature[12] and report[53]; $C$ represents the installed capacity of unit $r$; $LT$ represents the average campaign lifetime of coal power units(year); Age represents the operating time of unit $r$ in year $y$ (year). A 40-years campaign lifetime is presumed, consistent with the setting of Historical strategy. Retiring units that have operated for more than 40 years would not lead to any stranded assets, since initial capital costs are fully paid[54].

Shaped by unit-level heterogeneities, diverse decision orientations in China's coal power plant turnover (i.e., climate change mitigation, public health safeguard, economic loss avoiding) are taken into consideration in our study. To incorporate those policy preferences into phaseout decision making, we conduct a preference analysis based on multi-criteria decision-making (MCDM) method. We first normalize the cumulative economic benefits of decarbonization, cumulative health co-benefits, and cumulative assets stranding of all possible turnover in sequence by Min-Max Normalization method, following Eq. (14):

$$X_i^* = (X_i - X^{min})/(X^{max} - X^{min}) \qquad (14)$$

Then, the preference weighting factors (i.e., $\alpha$, $\beta$, and $\gamma$) are applied to represent the diverse decision preferences towards phaseout decision making. We calculate the normalized net benefits (dimensionless indicator) by taking the weighted sum of normalized cumulative economic benefits of decarbonization, normalized cumulative health co-benefits, and normalized cumulative assets stranding following Eq. (15):

$$Normalized\ net\ benefit_i = \alpha \times Benefit_{decarbonization,i}^* + \beta \times Benefit_{Health,i}^* \\ - \gamma \times Cost_{assets\,stranding,i}^* \qquad (15)$$

where the sum of $\alpha$, $\beta$ and $\gamma$ identically equals three. A lower value of normalized net benefits indicates a relatively lower cost effectiveness of specific turnover under certain decision preference.

## Data availability
The unit-level emissions and premature deaths data of coal power are compiled from CPED and available at Zenodo (https://doi.org/10.5281/zenodo.10672759). The raw CPED are protected and are not available due to data privacy laws. The Multi-resolution Emission Inventory of China database is available from http://www.meicmodel.org.cn/. The ground-level $PM_{2.5}$ concentration used in this study is available at http://tapdata.org.cn/. The baseline mortality incidence data are available at https://ghdx.healthdata.org/gbd-2019.

## Code availability
The code of the GEOS-Chem adjoint model to simulate the sensitivities of pollutant emissions to $PM_{2.5}$-related premature deaths is available at http://wiki.seas.harvard.edu/geos-chem/index.php/GEOS-Chem_Adjoint. The code used to manipulate the data and generate the results is available from the corresponding author upon reasonable request.

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

## Acknowledgements

This work is supported by the National Natural Science Foundation of China (72243008 and 72274106), the New Cornerstone Science Foundation through the Xplorer Prize, the Energy Foundation (G-2009-32416) and Tsinghua University Initiative Scientific Research Program (20223080041).

## Author contributions

D.T., Y.Lei, and Q.Z. designed the study. X.Y. performed model development and power fleet turnover projection with support from Y.Liu, Y.Z., J.C., Q.S., R.X., X.Q., C.C. and D.Z. on data compilation and from Y.Lei, S.C. and K.H. on analytical approaches. D.T. and X.Y. interpreted the data. X.Y., D.T., Y.Lei, and Q.Z. wrote the paper with input from all co-authors.

## Competing interests

The authors declare no competing interests.
