## [Peer Review File · Nature Communications]

Cost-effectiveness uncertainty may bias the decision of coal power transitions in ChinaREVIEWER COMMENTS

Reviewer #1 (Remarks to the Author):

Manuscript Number: NCOMMS-23-45720-T

Title: Cost-effectiveness uncertainty may bias the decision of coal power transition in China

Overall comment:

The paper modelled the cost-effectiveness uncertainty brought by policy implementation disturbance of different phaseout and new-built strategies (i.e., the disruption of phaseout priority) of coal power based on a developed unit-level uncertainty assessment framework, and revealed the opportunity and risk of coal transition decision by employing preference analysis. It argues that the uncertainty of policy implementation might lead to potential delays in yielding the initial positive annual net benefits. For example, a delay of 6 years might occur when the prior phaseout practice is implemented. In general, it is suggested that it can be accepted with modifications.

Specific comment 1:

The introduction section needs to provide an enriched background of this research and give a more convincing literature to support the view.

Specific comment 2:

In the results section, it is necessary to give a more detailed description which not just a description of the resulting figure.

Specific comment 3:

In how far are your assumptions and five scenarios in line with the announcements of the Chinese government aiming at climate neutrality by 2060? Coal power will still have about 0.5 GtCO₂ by 2060, inconsistent with China's energy transition envisioning the power sector becoming carbon-negative and providing emissions space for other industries.

Specific comment 4:

From the CO₂ emissions aspect, there are basically two emission pathways (Historical strategy and others), and deepening the decarbonization scenarios is not as effective in reducing emissions, which partly needs to be explained.

Specific comment 5:

Assumptions for all scenarios: the new-built units are randomly constructed on the site of retired units. But the actual situation is that China will be adding huge amounts of coal power in order to ensure the safe supply of electricity and flexible adjustment. Will the huge new coal plants (perhaps over 300GW) have an impact on the results?

Specific comment 6:

What is the reasoning behind choosing 40 years as lifetime of a coal power plants. Units in China often only expire at 25-30 years (In EU often only expire at 40-50 years). In how far does your result change if you assume 30 years?

Specific comment 7:

What are your assumptions regarding potential CCS deployment in China? Would retrofitting existing coal power units be possible and change the benefit value of coal power stranded assets?

Specific comment 8:

I assume that your database will also have some blank spots for some information. How do you deal with these items?

Specific comment 9:

Some work should be done to revise the writing style of the paper. At the moment it is a bit too prescriptive and has some stylistic issues. It will benefit from grammar and style correction from a native speaker.

Specific comment 10:

Other articles of interest for you might be

"Radical transformation pathway towards sustainable electricity via evolutionary steps." *Nature communications* 10.1 (2019): 1-16.

"Quantifying stranded assets of the coal-fired power in China under the Paris Agreement target." *Climate Policy* 23.1 (2023): 11-24.

"Estimating stranded coal assets in China's power sector." *Utilities Policy* 75 (2022): 101352.

"Coupling effects of cross-region power transmission and disruptive technologies on emission reduction in China." *Resources, Conservation and Recycling* 189 (2023): 106773.

Reviewer #2 (Remarks to the Author):

The authors explored the cost-effectiveness uncertainty brought by policy implementation disturbance of different phaseout and new-built strategies (i.e., the disruption of phaseout priority) of coal power based on a developed unit-level uncertainty assessment framework, and revealed the opportunity and risk of coal transition decision by employing preference analysis. Overall, the topic is interesting, timely and falls within the scope of this journal. I would like to take a further review and probably cite this article in my future work if the following concerns are well addressed.

1. Considering the demand of coal power also influences how coal power transition. How do the authors consider the impact of this effect on the result?
2. Whether then development of clean energy needs to be taken into account in evaluating the decision to phaseout and new construction.
3. A bit more background and mathematical description on death intensity definitions should be given.
4. The authors could provide more detail on the amount of data.
5. Which policy scenario is most appropriate for coal power transition in China, and how feasible is the policy in this scenario?

Reviewer #3 (Remarks to the Author):

The present paper dwells on the intricacies of transition from fossil fuel-based electricity generation to renewables in China. Recent evidences cited in the paper suggest that implementation of the 11th Five Year Plan led to closure of 127 GW of relatively inefficient coal-based Electricity Generating Units (EGUs), while at the same time more efficient EGUs are also slated to start production in the future. In this context, the work tries to derive compatible pathways to phaseout coal based EGUs.

There can be multiple criteria to determine the phaseout strategies. The authors adequately address them by considering five phaseout strategies: Historical strategy (allow plants to operate for their historical expected lifetime); BAU strategy (consistent with existing phase out strategy); Death strategy (prioritize public health protection); Carbon strategy (prioritize climate change mitigation) and Age-to-Capacity strategy (prioritize economic loss avoidance). After providing weights to different strategies, the authors use multiple criteria decision making (MCDM) method in an attempt to generate pareto-optimal solutions of strategic options suitable for China. Use of this methodology is appropriate to deal with the uncertainties in the model. The estimates with their associated confidence intervals are also reported in an adequate manner.

Notably, the authors use α as the weight for climate change mitigation strategy, β as the weight for health benefits and γ as weight for stranded asset avoidance. The authors suggest to use a “dimensionless indicator (named the normalized net benefit) obtained by normalizing and weighed summing the above three factors” to help the decision makers in strategy assessment based on their time-varying preferences. Obviously, the idea holds merit. May the authors be asked to demonstrate the workability of the idea by calculating the same on a limited dataset of a particular region of China?

In reference to the Death strategy, it may be noted that exposure to PM2.5 leading to mortality is calculated in this paper using the GEMM as developed by Burnett et al. The model may be using concentration response functions to estimate the relative risk of deaths due to non-communicable diseases and lower respiratory infections. Recent advances in the literature suggest that exposure to sulphur, sulphates or PM2.5 from coal emissions are associated to higher morbidity in comparison with PM2.5 from other sources per unit concentration. Recent studies (Henneman et al) shows that exposure to coal PM2.5 was associated with 2.1 times greater mortality risk then exposure to PM2.5 from all other sources. Maybe, the death estimates by the authors of this paper are biased downward in the light of the above revelation.

The present study is static in nature. A longitudinal study that considers the dynamic changes due to economic cycles may lead to alteration of ranking of power plants closures under the strategies considered in the paper. Economic downturns may lead to lower demand of electricity and thus lower production from coal plants. Introduction of renewables in the grid may also lead to altered production schedule and lesser usage of these coal based generating units. To address these issues, the authors may be encouraged to run the modelling exercise periodically, more specifically quinquennially in sync with the Five-year Plans of China.

The plan of the work is very well laid out and executed in a very comprehensive way.

Reference

Lucas Henneman et al., Mortality risk from United States coal electricity generation. Science 382, 941-946 (2023). DOI:10.1126/science.adf4915

(S.Majumdar)

REVIEWER COMMENTS

Reviewer #1 (Remarks to the Author):

Overall comment:

The paper modelled the cost-effectiveness uncertainty brought by policy implementation disturbance of different phaseout and new-built strategies (i.e., the disruption of phaseout priority) of coal power based on a developed unit-level uncertainty assessment framework, and revealed the opportunity and risk of coal transition decision by employing preference analysis. It argues that the uncertainty of policy implementation might lead to potential delays in yielding the initial positive annual net benefits. For example, a delay of 6 years might occur when the prior phaseout practice is implemented. In general, it is suggested that it can be accepted with modifications.

Response: We thank the Referee for the positive and accurate summary of our work and for the fair and thoughtful comments below. We've added new text and data materials, made a number of revisions in response, and believe the manuscript has been substantially improved. A point-to-point response is presented below.

Specific comment 1:

The introduction section needs to provide an enriched background of this research and give a more convincing literature to support the view.

Response: We thank the Referee for the constructive comment. The revised Introduction section has provided an enriched background by adding a comprehensive description on the potential challenges and trends related to future coal power transition. For example, a steady but evolutionary transition planning in Chinese coal power is still difficult and uncertain (Bogdanov et al, 2019), as the majority of existing units were introduced around 2010 and might phaseout in a similar timeframe (Zhang et al., 2022). Current phaseout decision has gone beyond the scope of pollution control, necessitating further consideration of public health and climate mitigative (Zhang et al., 2021; Zhuo et al., 2022). The absence of quantifying the patterns and uncertainties inherent in prior phaseout practices would bias the cost-effectiveness estimation of coal transition. Therefore, it is essential to retrospect on prior phaseout policy and consider the main factors of costs, climate impacts, and health burdens in shaping future turnover.

Also, the suggested references by the Referee and other studies have been added as demonstrations to bolster the persuasiveness of our standpoint.

References:

Bogdanov, D. *et al.* Radical transformation pathway towards sustainable electricity via evolutionary steps. *Nature Communications* **10**, 1077 (2019). <https://doi.org/10.1038/s41467-019-08855-1>

Zhang, W., Ren, M., Kang, J., Zhou, Y. & Yuan, J. Estimating stranded coal assets in China's power sector. *Utilities Policy* **75**, 101352 (2022). <https://doi.org/https://doi.org/10.1016/j.jup.2022.101352>

Zhang, S. H. *et al.* Incorporating health co-benefits into technology pathways to achieve China's 2060 carbon neutrality goal: a modelling study. *Lancet Planet Health* **5**, E808-E817 (2021). [https://doi.org/10.1016/S2542-5196\(21\)00252-7](https://doi.org/10.1016/S2542-5196(21)00252-7)

Zhuo, Z. *et al.* Cost increase in the electricity supply to achieve carbon neutrality in China. *Nature Communications* **13**, 3172 (2022). <https://doi.org/10.1038/s41467-022-30747-0>

Specific comment 2:

In the results section, it is necessary to give a more detailed description which not just a description of the resulting figure.

Response: We thank the Referee for the constructive suggestions. We have added a more detailed description behind the figures as suggested. On the one hand, numerous sensitivity tests are carried out and added into the revised Result section, in order to explore potential variations related to phaseout factors and provide a more comprehensive understanding of coal power transition, as well as associated decisions. On the other hand, we have gone beyond the description of resulting figures and reorganized the Result section, by adding a thorough interpretation of the difference in outcomes across diverse phaseout pathways, based on an in-depth analysis and comparative sensitivity tests.

Specific comment 3:

In how far are your assumptions and five scenarios in line with the announcements of the Chinese government aiming at climate neutrality by 2060? Coal power will still have about 0.5 GtCO₂ by 2060, inconsistent with China's energy transition envisioning the power sector becoming carbon-negative and providing emissions space for other industries.

Response: We thank the Referee for the valuable and thoughtful comment. We totally agree that the power sector aiming at climate neutrality by 2060 is likely to be carbon-negative, providing emissions space for other industries via the increasing penetration of bioenergy with carbon capture and storage (BECCS; Tang *et al.*, 2021, Fan *et al.*, 2023).

For this study, we directly selected the coal power demand trend of *On-time peak-net zero-clean air* scenario designed by Cheng *et al.*, 2023, which can achieve climate neutrality by 2060 with totally 0.9 Gt CO₂ emissions from all sectors. We apologized for neglecting CCUS retrofit when power's decarbonization through phaseout decision, leading to an overestimation of CO₂ emission trajectories. During this revision, we have corrected the assumption of coal power demand by considering the penetration rate of CCUS from Cheng *et al.*, 2021. As a result (as shown in new Figure 2), CO₂ emissions of coal power would drop to 31-36 Mt (i.e., near zero) by 2060, and such emissions would probably been offset by BECCS to achieve carbon-negative in the power sector, which has been discussed in the revised main text (Line 113-Line 118).

Meanwhile, the detailed methodology of how CCUS retrofit has been added in Method section (Line 504-Line 527).

Trends and uncertainties of different coal power transition strategies

“CO₂ emissions of coal power would peak at 4.1–4.3 Gt in 2030 and drop to 31–36 Mt (i.e., near zero, ~3% of total national emissions²⁴) by 2060 (Fig. 2a), as the demand of coal power has drop to 8% of current level and the majority of coal power units are projected to complete CCUS retrofitting (~99%; see Supplementary Figs. 3 and 4). Such emissions from coal power would probably been offset by bioenergy with carbon capture and storage (BECCS) to achieve carbon-negative in the power sector^{25,26}. ”

CCUS retrofitting.

“A method framework of CCUS retrofit in Chinese coal-fired power units is shown in Supplementary Fig. 21. We first combine the future harmonized coal power demand projection and the CCUS retrofitting ratio (see Supplementary Fig. 4) explored by Cheng et al., 2021²⁴ to obtain possible supply curve of provincial CCUS retrofit in coal power. Most provinces are slated to initiate large-scale commercialization of CCUS around 2030, with the majority of coal capacity completing retrofitting by 2060.

Similar to the phaseout algorithm (see Modeling future possible turnover section), a CCUS retrofitting algorithm is further designed to retrofit a certain coal power capacity within each province in China for a given year, according to each possible phaseout priority of specific strategy and unit-level generation projection. To minimize the risk of stranding assets resulting from CCUS retrofitting, we assume that the installation requirements for CCUS would be prioritized for new-built capacity. If the new-built units could not meet the supply curve of Coal-CCUS demand, old units with less phaseout priority are prioritized for CCUS retrofitting to fill the gap. A sensitive test of the allocation of CCUS installation requirements among new-built and existing capacities is further discussed in Supplementary Text 5.

Although CCUS could significantly reduce CO₂ emissions, the additional electricity demand as CCUS systems in the future may have a non-negligible impact on fuel consumption and coal-related health risk^{43,44}. In this study, coal power units with CCUS installation are adopted to achieve a CO₂ capture rate of 90% with an additional 15% energy consumption. Furthermore, it is reported that the unit capital cost of CCUS retrofitting would vary from 3300 RMB/kW to 8500 RMB/kW^{27,45}. A potential life extension of 15 years is allocated to units with CCUS installation, aligning with retrofitting investment recovery periods that takes the high cost of capital cost, operating and maintenance cost into consideration^{27,28}. The potential impact of life extension is then feededback to adjust the Coal-CCUS demand gap.

”

Figure 2 | Trends and possible ranges of CO₂ emissions, deaths, and benefits for each strategy. (a) CO₂ emissions. (b) Coal-power-related premature deaths. (c) Monetized benefits compared to the historical phaseout pathway.

Supplementary Figure 21 | Method framework of CCUS retrofit in coal-fired power units.

Supplementary Figure 4 | Penetration rate of Carbon Capture, Utilization, and Storage (CCUS) in coal-fired power units.

References:

Cheng, J. *et al.* A synergistic approach to air pollution control and carbon neutrality in China can avoid millions of premature deaths annually by 2060. *One Earth* **6**, 978-989 (2023). <https://doi.org/10.1016/j.oneear.2023.07.007>

Cheng, J. *et al.* Pathways of China's PM_{2.5} air quality 2015–2060 in the context of carbon neutrality. *National Science Review* **8** (2021). <https://doi.org/10.1093/nsr/nwab078>

Fan, J.L. *et al.* Co-firing plants with retrofitted carbon capture and storage for power-sector emissions mitigation. *Nat. Clim. Chang.* **13**, 807–815 (2023). <https://doi.org/10.1038/s41558-023-01736-y>

Tang, H. T., Zhang, S. & Chen, W. Y. Assessing Representative CCUS Layouts for China's Power Sector toward Carbon Neutrality. *Environmental Science & Technology* **55**, 11225-11235 (2021). <https://doi.org/10.1021/acs.est.1c03401>

Specific comment 4:

From the CO₂ emissions aspect, there are basically two emission pathways (Historical strategy and others), and deepening the decarbonization scenarios is not as effective in reducing emissions, which partly needs to be explained.

Response: We thank the Referee for the constructive comment. The detailed explanation of why CO₂ mitigation potential of Climate-targeted strategy is not so obviously effective compared to Historical strategy has been supplemented as suggested.

On one hand, the differences of CO₂ emissions among different strategies are only from the phasing out process of existing power units as the same assumption of energy efficiencies for new-built power units. Specifically, all the new-built capacities will be equipped with the same

ultra-supercritical combustion technology by ignoring the unit-level heterogeneity in energy efficiency (i.e., 270 gce/kWh referencing NDRC, 2021; GB 21258-2017).

On the other hand, compared to emission intensity of air pollutants (defined as emission per installed capacity) across several magnitudes (Tong et al., 2018; Tang et al., 2019), the distribution of unit-level emission intensity of CO₂ exhibits a relatively even pattern (see Supplementary Fig. 1). Additionally, part of low-efficient power units with higher CO₂ emission intensities will be identified in all strategies. Therefore, the effectiveness of decarbonization may not be distinctive when phasing out the same amount of installed capacity. A description about “the limited decarbonization effectiveness” has been added in the revised Result as suggested and shown below (Line 118-Line 126).

In order to display the differentiated cumulative effects of decarbonization, we design a baseline test, where the energy efficiency of new capacities are assumed to be ~308 gce/kWh within Historical strategy, consistent with the average efficiency of coal power in 2018. For all targeted strategies, the effective of decarbonization is more obvious in the baseline test compared to the original scenario (see Supplementary Fig. 8a), due to the consideration of the advancement in combustion technology (i.e., 270 gce/kWh). Especially in the climate-targeted Carbon strategy, the cumulative CO₂ emission reduction amounts to 7.1 Gt, around twice the annual emissions of coal power in 2018. As a result, all targeted strategy can achieve a positive outcome in the baseline test, due to the amplification of the cumulative effects of decarbonization and health protection from technological advancement (see Supplementary Fig. 8b). The “baseline tests for energy efficiency of new-built capacity” are analyzed in detail in revised Result (Line 169-Line 175) and Supplementary Text 2 as shown below.

Trends and uncertainties of different coal power transition strategies

“The differences in emission trends among strategies are relatively minor, with variations of less than 5% in 2030, due to the successful identification of part of low-efficient power units with higher CO₂ emission intensities in all strategies (Supplementary Fig. 1) and the same assumption of new capacity by ignoring the unit-level heterogeneity (i.e., 270 gce/kWh in energy efficiency, see Methods). Nevertheless, Carbon strategy would hold in a most effective CO₂ emission reduction of 210 Mt compared to Historical strategy in 2030, which is around 1.4 times greater than that achieved through Age-to-Capacity strategy. It indicates that the strategic phaseout decision is still effective for deeper decarbonization.”

Cumulative cost-benefits and uncertainties

“Taking the advancements in combustion technology driven by phaseout into consideration, the cumulative decarbonization efforts are further amplified among all targeted strategies (Supplementary Text 2; Supplementary Fig. 8), if Historical strategy remains current average energy efficiency (i.e., 307.6 gce/kWh). Especially in the Carbon strategy, CO₂ emissions are cumulatively reduced by over 7 Gt during 2018 to 2060. It indicates that both strategic phaseout and energy efficiency improvements would make a difference in decarbonization.”

Supplementary Text 2. Baseline test for energy efficiency of new-built capacity

“Assessing the potential impact of energy efficiency of new-built capacity is necessary in our study, enabling an analysis of how energy efficiency improvements affect the benefit of targeted phaseout strategies. On one hand, new-built units will gradually take the lead after 2040, assuming average lifetime of 30-40 years for all existing coal power units. On the other hand, proposed coal power projects are subject to more stringent clean requirements since 14th Five-Year Plan¹. Quantitative requirements of ultra-supercritical units with coal consumption rate lower than 270 gce/kWh are recommended, which is more advanced than the energy efficiency of current coal power fleet in 2018 (i.e., 307.6 gce/kWh).

Supplementary Fig. 8 shows that the effect of decarbonization is more obvious across all targeted strategies, when taking the advancement in combustion technology into consideration, by using a Historical strategy where new-built capacities would operate at the current energy efficiency level as benchmark. Carbon strategies, which is designed for decarbonization, can reduce over 7 Gt emissions during 2018-2060. This amount is equivalent to the emissions produced by coal power over a two-year period.

In this baseline test, the net benefits of all targeted strategies would significantly increase. For example, the cumulative net benefit of Carbon strategy increased from -378 (CI, -527~-231) to 215 (CI, 65~362) billion RMB if the energy efficiency of new-built capacity in Historical is set to 307.6 gce/kWh. This is due to replacing backward generating units with other inefficient units would not result in a swifter pace of efficiency improvement. It might restrict the potential benefits associated with decarbonization and mortality reduction within Historical strategy of baseline test. To prevent falling into a new cycle of technological lock-in and mitigate the risk of overall negative benefit, it is imperative to replace outdated generating units with efficient ones in a cost-effective manner.”

Supplementary Figure 8 | Comparison of the cumulative CO₂ emission reduction and net benefits under original scenario and baseline test considering technological advancement.

Supplementary Figure 1 | The distribution of unit-level emission intensity. Points represent individual coal-fired units, in each case plotted according to nameplate capacity (y-axis) and annual emissions of SO₂ (a), NO_x (b), PM_{2.5} (c) and CO₂ (d) (x-axis). Solid diagonal lines indicate the median emission intensity (tonnes emission per MW) and shaded triangles indicate units whose emission intensity is over 90th percentile.

References:

Tong, D. et al. Targeted emission reductions from global super-polluting power plant units. *Nat. Sustain.* 1, 59-68 (2018).

Tang, L. et al. Substantial emission reductions from Chinese power plants after the introduction of ultra-low emissions standards. *Nature Energy* 4, 929-938 (2019).

National Development and Reform Commission & National Energy Administration. Implementation Plan for National Coal Power Unit Renovation and Upgrading (2021) (in Chinese).

General Administration of Quality Supervision, Inspection and Quarantine of the People's Republic of China. The norm of energy consumption per unit product of general coal-fired power set (GB 21258-2017) (in Chinese).

Specific comment 5:

Assumptions for all scenarios: the new-built units are randomly constructed on the site of retired units. But the actual situation is that China will be adding huge amounts of coal power in order to ensure the safe supply of electricity and flexible adjustment. Will the huge new coal plants (perhaps over 300GW) have an impact on the results?

Response: We thank the Referee for the thoughtful comment. We totally agree that adding huge amounts of coal power is to ensure the safe supply of electricity and flexible adjustment, and new-built coal power capacity does have a great impact on the results. Detailed description of the impact of new-built capacity has been supplemented as suggested.

The growth trend of coal power is anticipated to persist until the attainment of carbon peaking in 2030, taking into account factors such as economic growth, electrification and structural change in the industry (Zhang et al., 2022; Zhuo et al., 2022; Yu et al., 2022). A large number of new-built units accounting for ~700 GW by 2030 is required to address the potential power generation gap, if meanwhile the annual utilization hours drop at 2% rate per year as recently.

Supplementary Fig. 7 shows the significant influence of new-built capacity on the future power fleet. Generation attributed to new-built units is up to 2,500 TWh (45% of total generation) in Historical strategy and over 3,600 TWh (65%-70% of total generation) in all targeted strategies in 2030. Over 60% of CO₂ emissions and health risk are attributed to new-built capacity within all strategies. Additionally, new-built capacity contributes 40%-50% of uncertainty in annual net benefits of the phaseout decision before 2040 (Supplementary Fig. 6). For Death strategy, the contribution increases to around 80% by 2050, since the majority of existing units has been phased out. This highlights the importance of selecting suitable sites for new plants to mitigate health risks and alleviate uncertainties.

Even average annual utilization hour is stable in future (i.e., not change from 2018-2060), conduct by removing the assumption of a year-by-year decrease in capacity factor, new-built units still account for over 300 GW by 2030. Therefore, the amount of new-built coal power capacity does have a great impact on the results. Our tests show that the net benefit for all targeted strategies would further amplify, if keeping the original annual utilization hours (Supplementary Fig. 12). This is because the existing units with lower phaseout priorities and better performance contributes more to the coal power generation (40%~45% in 2030), compared to the original scenario. However, the preference-based phaseout decision shows limited changes compared to the original scenario (Supplementary Fig. 23), because the gap of normalized benefit and gap remains still in both scenarios.

Detailed description of the impact of new-built capacity are added in revised Result (Line 152-Line 158) and Supplementary Text 4 as shown below.

Trends and uncertainties of different coal power transition strategies

“While, as the majority of existing units were phased out around 2050, the increasingly dominate role of new-built capacity in supply future coal power generation within all targeted strategies (from ~65% in 2030 to ~85% in 2040), especially under the anticipate of reductions in utilization rate of both existing and new capacity (i.e., more new-built capacity is needed to

supply the same generation demand; Supplementary Text 1; Supplementary Fig. 7), reveals that new-built capacity does hold significance in shaping net benefit and its associated uncertainty.”

Supplementary Text 1. The potential impact of new-built capacity

“Supplementary Figure 7 shows the trends in coal power generation, deaths, and CO₂ emissions attributed to new-built capacity for each strategy. For all strategies, the overwhelming magnitude of new-built coal power capacity does hold significance in shaping the future power fleet, with distinct roles played by new-built capacity in targeted strategies compared to the Historical strategy. Across all targeted strategies, new-built units, accounting for with 65-67% of total installed capacity, would contribute to over 3700 TWh coal power generation and 2.5 Gt CO₂ emissions in 2030. Meanwhile, new-built units would only account for 45% of total installed capacity with 1.7 Gt CO₂ emissions when implementing Historical strategy, due to the slower phaseout pace.

New-built units are expected to dominate coal power fleet after 2040, as the majority of existing units will have been phased out in targeted strategy. For example, nearly 90% of coal power generation is generated by new-built units in all targeted strategies by 2045, contrasting with a 48% dependence on new-built units within Historical strategies. The slower phaseout pace in Historical strategy has prompted increasing CCUS retrofitting in existing units, enabling continuous operation beyond their 40-year lifetime.

Supplementary Figure 6 shows the contributions of phaseout of existing units and construction of new-built units to the uncertainty of annual net benefits of each strategy. The construction of new units contributes to a certain level of uncertainty in net benefits of phaseout decision. For Death strategy, 40%-50% of the total benefit uncertainty of phaseout benefit could be attributed to the new-built power units before 2040. The contribution of new-capacity keeps increasing to around 80% in 2050. This is mainly due to the uncertain selection of sites for new-built capacity has a significant impact on power-related health risks' uncertainty. Thus, choosing suitable site for constructing new coal-fired power plants can effectively mitigate the health risks associated with coal power and alleviate the risks brought about by uncertainty.”

Supplementary Text 4. Sensitivity test for annual utilization hours

“In recent years, the annual utilization hours of coal power have been declining brought out from overcapacity⁷. It might continue to decrease in the future with the call for flexibility transformation. Given diverse trajectories, uncertainty exists in forecasting the operating hours of coal-fired power plants. Sensitivity test for annual utilization hours is added by removing the assumption of a year-by-year decrease in capacity factor.

Supplementary Fig. 12 shows that the net benefit for all targeted strategy would further amplify, if keep the original annual utilization hours. For example, the net benefit of Carbon strategy would increase to 95 (CI, -197~275). The probability to achieve a positive outcome is around 70%. This is because over 40% of generation are attributed to existing units with lower phaseout priorities and better performance in 2030, if their annual utilization hours are not forced to decrease. When it comes to preference-based phaseout decision, limited changes compared to the original scenario (Supplementary Fig. 23), because the gap of normalized benefit and cost remains stable in both scenarios.”

Supplementary Figure 7 | Trends of coal generation, deaths, CO₂ emissions of new-built capacity across each strategy with median net benefit. (a) Coal power generation. (b) CO₂ emissions. (c) Coal power-related premature deaths.

Supplementary Figure 6 | Decomposition of uncertainty in net benefits trend of each coal power transition strategy.

Supplementary Figure 12 | Comparison of the cumulative net benefits under original scenario and sensitivity test assuming stable annual utilization hours for all units.

Supplementary Figure 23 | Preference analysis for coal power phaseout strategy design. Coal power phaseout strategy selection according to the maximum normalized total benefits under original scenario (a) and sensitivity test assuming stable annual utilization hours for all units (b). α , β , γ represents the preference weighting factor of climate change mitigation, public health protection and assets stranding avoiding, respectively. The higher value of the specific preference weighting factor, the more emphasis is placed on that preference.

References:

Zhang, S. & Chen, W. Y. China's Energy Transition Pathway in a Carbon Neutral Vision. *Engineering-Proc* **14**, 64-76 (2022). <https://doi.org/10.1016/j.eng.2021.09.004>

Zhuo, Z. Y. *et al.* Cost increase in the electricity supply to achieve carbon neutrality in China. *Nature Communications* **13** (2022). <https://doi.org/ARTN317210.1038/s41467-022-30747-0>

Yu, S. *et al.* Synthesis Report 2022 on China's Carbon Neutrality: Electrification in China's Carbon Neutrality Pathways. (2022).

Zhang, J.; Li, X.; Pan, L. Policy Effect on Clean Coal-Fired Power Development in China. *Energies* **2022**, *15*, 897. <https://doi.org/10.3390/en15030897>

Specific comment 6:

What is the reasoning behind choosing 40 years as lifetime of a coal power plants. Units in China often only expire at 25-30 years (In EU often only expire at 40-50 years). In how far does your result change if you assume 30 years?

Response: We thank the Referee for pointing out this very important aspect. The lifetime of coal power plants is a very important factor in deciding the coal power phaseout and associated cost-effectiveness, which is expressed in both technical and economic terms (IEA, 2013). The adoption of 40 years as lifetime of coal power plants is mainly based on the knowledge of global average lifetimes (37 years in Davis et al., 2014) and assumptions in scenarios from previous studies (Tong et al., 2019; Jewell et al., 2019; Wang et al., 2020). Therefore, for this study, a 40-year lifetime for Historical strategy is assumed as the baseline of coal fleet turnover.

We totally agree that units in China often expire less than 40 years (Jiang et al., 2017), which has been well demonstrated since the 11th Five-Year Plan (the average lifetime for the retired ones was only 23.1 years). Therefore, for all targeted strategies in this study (i.e., BAU, Death, Age-to-Capacity and Carbon strategies), the assumption of all existing units shut down by 2050 has been applied. It is estimated that the lifetime is around 25.8 years on average. A more detailed description of the strategy design together with the assumption of lifetime and phaseout rate has been added in Introduction section (Line 71-Line 80).

Meanwhile, we have added a sensitivity test by accelerating the phaseout pace to evaluate the potential impact of the future phaseout rate, specifically, setting 30 years lifetime in Historical strategy as suggested and assuming phaseout all existing units by 2040 for all targeted strategies (~21.2 years lifetimes on average). Results shows that the risk of cumulative negative outcomes significantly increases with a faster introduction of new-built capacity (Supplementary Fig. 10), diminishing the relative cumulative benefits compared to the Historical strategy. For preference-based decision making, Age-to-Capacity strategy favored as the optimal strategy in more preferences for its advantage in asset stranding avoidance (Supplementary Fig. 22), within a faster phaseout pace. This will in turn emphasize the importance of minimizing policy disturbances associated with targeted phaseout strategy and reducing the risks of negative benefit. The sensitivity tests are analyzed in detail in revised Result section (Line 188-Line 192) and Supplementary Text 3 as shown below.

Introduction

“Under the harmonized provincial coal power demand projections²³ and the penetrations of Carbon Capture, Utilization and Storage (CCUS) in line with carbon peak and carbon neutrality goals^{23,24}, we defined five phaseout strategies (see Extended Data Table 1 in detail): the first allows power plants to operate for their historical expected lifetime (i.e., Historical strategy; assuming 40 years lifetime); the second remains consistent with the previous phaseout practice (i.e., business as usual, BAU strategy); the third gives priority to public health

protection (i.e., Death strategy); the fourth targets climate change mitigation (i.e., Carbon strategy); and the final strategy targets economic loss avoidance (i.e., Age-to-Capacity strategy). Within strategies other than Historical, we assumed all existing coal capacity would be shut down by 2050 (~25.8 years of lifetime).”

Cumulative cost-benefits and uncertainties

“When it comes to a faster phaseout rate to close the prior practice (i.e., 30-year lifetime for Historical strategy and assuming an entire phaseout of existing units by 2040 for all targeted strategies), the risk of negative outcomes would be amplified by rapidly diminishing the benefit gap between Historical and other strategies (Supplementary Text 3; Supplementary Fig. 10).”

Supplementary Text 3. Sensitivity test of future phaseout rate

“The decision to phaseout a coal-fired power unit is usually driven by economic factors of operating costs, replacement costs, and revenues². Globally, coal plants have retired at an average lifetime of over 45 years, thus, a lifetime of 40 years for power plant is widely adopted within the research community³⁻⁵. However, it is worth noting that China has actively mandated the replacement of outdated generating capacity with more advanced and younger power plants since the 11th five-year plan, resulting in a shorter operational lifetime⁶. A sensitivity test is conducted to evaluate how changes in phaseout rate would affect the cost-effectiveness of the phaseout decision.

Supplementary Fig. 10 shows the comparisons of cumulative cost-benefits and uncertainties for each strategy between the original scenario (setting a lifetime of 40 years for Historical strategy and a faster retirement rate of 40% in 2030 and 100% in 2050 for other strategies) and a sensitivity test with faster phaseout rate (setting a lifetime of 30 years for Historical strategy and a faster retirement rate of, 40% in 2025 and 100% in 2040 for other strategies). The risk of cumulative negative outcomes significantly increases. For example, the probability of Age-to-Capacity strategy experiencing cumulative negative outcomes rises to over 80%. This is because an accelerated phaseout rate for all strategies would diminish the relative cumulative benefits of health and decarbonization of targeted strategies compared to the Historical strategy, by a more rapid introduction of new-built capacity. Supplementary Fig. 22 result shows Age-to-Capacity strategy would favor as the optimal strategy in more preferences for its advantage in asset stranding avoidance. This is due to the relative gap in normalized asset stranding is further amplified, within a faster phaseout pace. It is essential to reduce the policy implementation disruptions and minimize uncertainties, especially in light of the heightened risk of negative benefits.

”

Supplementary Figure 10 | Comparison of the cumulative net benefits under original and accelerated phaseout rate.

Supplementary Figure 22 | Preference analysis for coal power phaseout strategy design. Coal power phaseout strategy selection according to the maximum normalized total benefits under original phaseout rate (a) and accelerated phaseout rate (b). α , β , γ represents the preference weighting factor of climate change mitigation, public health protection and assets stranding avoiding, respectively. The higher value of the specific preference weighting factor, the more emphasis is placed on the that preference.

References:

IEA (2013), World Energy Outlook 2013, IEA, Paris <https://www.iea.org/reports/world-energy-outlook-2013>, License: CC BY 4.0

Davis, S. J. & Socolow, R. H. Commitment accounting of CO₂ emissions. *Environmental Research Letters* 9 (2014). <https://doi.org/Artn08401810.1088/1748-9326/9/8/084018>

Tong, D., Zhang, Q., Zheng, Y. *et al.* Committed emissions from existing energy infrastructure jeopardize 1.5 °C climate target. *Nature* **572**, 373–377 (2019). <https://doi.org/10.1038/s41586-019-1364-3>

Jewell, J., Vinichenko, V., Nacke, L. *et al.* Prospects for powering past coal. *Nat. Clim. Chang.* **9**, 592–597 (2019). <https://doi.org/10.1038/s41558-019-0509-6>

Wang, H. *et al.* Early transformation of the Chinese power sector to avoid additional coal lock-in. *Environmental Research Letters* **15**, 024007 (2020). <https://doi.org/10.1088/1748-9326/ab5d99>

Jiang, S., Chen, Z., Shan, L., Chen, X. & Wang, H. Committed CO₂ emissions of China's coal-fired power generators from 1993 to 2013. *Energy Policy* **104**, 295-302 (2017). [https://doi.org:https://doi.org/10.1016/j.enpol.2017.02.002](https://doi.org/https://doi.org/10.1016/j.enpol.2017.02.002)

Specific comment 7:

What are your assumptions regarding potential CCS deployment in China? Would retrofitting existing coal power units be possible and change the benefit value of coal power stranded assets?

Response: We thank the Referee for this constructive comment. we apologized for neglecting the CCS retrofit to only focus on the decarbonization brought by phaseout and new-built decision. During this revision, we have incorporated a CCUS retrofitting algorithm into the modelling framework to take potential CCUS deployment into consideration (Supplementary Fig. 21). A GCAM-based CCS retrofitting ratio explored by Cheng *et al.*, 2021 is applied to obtain supply curve of provincial CCUS retrofitting in coal power. Most provinces are expected to drive the commercialization of CCUS around 2030, with the majority of coal capacity projected to complete retrofitting by 2060 (see Supplementary Fig. 4). Units after CCUS retrofitting are set to achieve a 90% CO₂ capture rate with a 15% increase in energy consumption (EEA, 2011).

Considering additional capital investment in CCUS retrofitting, we assumed that new-built capacity would be assigned for the priority of CCUS deployment, and a 15-year term of lifespan protection (i.e., investment recovery periods considering capital cost, operating cost and revenues) is applied to pay off the cost of CCUS installation (Zhang *et al.*, 2014; Fan *et al.*, 2018). The revised results show it will not significantly affect the main results of the study although the introduction of CCUS retrofitting indeed have some impact on possible trend of emissions and health. A method framework for CCUS retrofitting has been added to Method section (detailed in response of comment 3).

Meanwhile, as the assumption of the priority of new-built units to install CCUS deployment, we have added a sensitivity test by allocating 50% of the CCUS demand to new-built units and assign the remaining demand to existing units. In this scenario, 255–270 GW of existing units are retrofitted and keep operating within targeted strategies. Results shows that the net benefits significantly increases if some advanced existing units are involved into CCUS retrofitting (Supplementary Fig. 13), which highlights the importance of strategic CCUS retrofitting among

existing coal power units. The sensitivity tests are analyzed in detail in revised Discussion section (Line 262-Line 272) and Supplementary Text 5.

Discussion

“It is noted that the tests of CCUS priority (i.e., allocating 50% of the CCUS demand to new-built units and assign the remaining demand to existing units) and stable annual utilization hours (i.e., remaining the current level annual utilization hours) show the net benefits of targeted phaseout would increase within all strategies (see Supplementary Texts 4 and 5). It’s because well-performing existing units with lower phaseout priorities have a chance to retain their roles in the mid- or long-term. Especially in the test of CCUS priority, more than 250 GW of existing units could be involved into CCUS retrofitting, and keep operating beyond their assumed lifetimes (15-year protection as investment recovery period of CCUS retrofitting^{27,28}). Those units would continuously expand the disparities in cost-effectiveness among strategies tailored to different preferences and ultimately make strategies other than BAU becomes more competitive across preferences (Supplementary Fig. 16). ”

Supplementary Text 5. Sensitivity test of CCUS priority

“Supplementary Fig. 13 shows a suitable allocation of CCUS installation among new-built and existing units may increase the net benefit of phaseout policy. For example, the median cumulative net benefits of Age-to-Capacity and BAU increase to 451(CI, 292~606) and 541 (CI, 372~707) billion RMB, when allocating 50% of the CCUS demand to new-built units and assigning the remaining demand to existing units. This is because more than 250 GW of existing units with lower phaseout priorities, which might have better performance (e.g., equip with better emission control technology) than new-built units, are involved into CCUS retrofitting. Extending the operational lifespan of those advanced units can contribute to reducing both health risks or CO₂ emissions associated with coal power, which highlights the significance of strategically retrofitting advanced coal power units with CCUS. Hence, Age-to-Capacity, Death and Carbon strategies have the potential to confer a competitive edge in preference-based decision-making compared to BAU strategy (Supplementary Fig. 16).”

Supplementary Figure 13 | Comparison of the cumulative net benefits under original scenario (i.e., CCUS would be prioritized for new-built capacity) and sensitivity test (allocating 50% of the CCUS demand to new-built units and assign the remaining demand to existing units).

Supplementary Figure 16 | Preference analysis for coal power phaseout strategy design. Coal power phaseout strategy selection according to the maximum normalized total benefits original scenario (i.e., CCUS would be prioritized for new-built capacity, **a**) and sensitivity test (allocating 50% of the CCUS demand to new-built units and assign the remaining demand to existing units, **b**). α , β , γ represents the preference weighting factor of climate change mitigation, public health protection and assets stranding avoiding, respectively. The higher value of the specific preference weighting factor, the more emphasis is placed on the that preference.

References:

Cheng, J. *et al.* Pathways of China's PM_{2.5} air quality 2015–2060 in the context of carbon neutrality. *National Science Review* **8** (2021). <https://doi.org/10.1093/nsr/nwab078>

European Environment Agency. *Carbon capture and storage could also impact air pollution*, <https://www.eea.europa.eu/highlights/carbon-capture-and-storage-could> (2011).

Zhang, X. *et al.* A novel modeling based real option approach for CCS investment evaluation under multiple uncertainties. *Applied Energy* **113**, 1059-1067 (2014). <https://doi.org/https://doi.org/10.1016/j.apenergy.2013.08.047>

Fan, J.-L., Xu, M., Li, F., Yang, L. & Zhang, X. Carbon capture and storage (CCS) retrofit potential of coal-fired power plants in China: The technology lock-in and cost optimization perspective. *Applied Energy* **229**, 326-334 (2018). <https://doi.org/https://doi.org/10.1016/j.apenergy.2018.07.117>

Specific comment 8:

I assume that your database will also have some blank spots for some information. How do you deal with these items?

Response: We thank the Referee for this thoughtful comment. Our unit-level power plant database does have some blank information (see Supplementary Table 3), which are filled in several steps. Unit-level information used in our study includes basic information (i.e., installed capacity, operating time), operating conditions (i.e., capacity factor, generation, coal consumption), emission information (emission factor, pollutants control rate). The blank spots typically occur in unit-level operating conditions and emission information. To ensure data integrity and preserve key patterns, we have established a comprehensive preprocessing scheme for missing values imputation for the baseline unit-level coal power emission inventory.

For blank spots of basic information and operating conditions, the imputation begins at the unit or plant level, where missing values are filled using prior data from the same units or plants. If not feasible, missing values are filled using data from units with similar installed capacity in the same province. The imputation is further validated according to the data type based on sectoral statistical data. Taking operating conditions as an example, the summation of unit-level coal power generation is cross-validated against provincial-level generation. The imputation for emission factors is similar to the above process, but the provincial variations in nature of fossil coal (e.g., sulfur content and ash content) is taken into account.

For blank spots of pollutants control rate, a dynamic model was developed to conduct the imputation, considering emission standard and retrofitting progress. For units with prior end-of-pipe control information, missing values are filled using prior data from the same units, if end-of-pipe control information has already met current emission standard. For other missing values, the emission standard is used to calculate the pollutants control rate, and the installation end-of-pipe device is prioritized for young units with large installed capacity. The model would iterate and self-correct until it satisfies the principle that end-of-pipe control information (i.e., retrofitting progress and average control level) after imputation aligns with statistical data.

Discussion of blank spots in our baseline emission dataset and description about “Preprocessing of baseline emission inventory of coal power” have been added in the revised Discussion section (Line 288-Line 291) and Supplementary Text 9.

Discussion

“Second, baselined emission dataset (i.e., CPED) contains a small number of blank spots and missing information (Supplementary Table 3) and it can comprehensively capture the key features of coal power fleet in China after appropriate imputation (Supplementary Texts 8 and 9).”

Supplementary Text 9. Preprocessing of baseline emission inventory of coal power

“Our unit-level power plant database (i.e., CPED) does have some blank information, which typically occur in unit-level operating conditions and emission information. Supplementary Table 3 shows the quantity of blank spots in the original emission database. To ensure data integrity and preserve key patterns, we have established a comprehensive preprocessing scheme for baseline unit-level coal power emission dataset. A step-wised method is designed to impute the missing value of activity-rate parameters, emission factors and pollutant removal efficiency.

For missing basic information and operating conditions, the imputation starts at unit- or plant-level where missing values are filled by the prior values from the same units or units in the same plant. If the first step is not feasible, average data from other units with similar installed capacity in the same province are used to fill the blank spots. Validation of imputed values is conducted based on the data type, which leveraging sectoral official statistical data. For example, the summation of unit-level coal power generation is cross-validated against provincial-level generation.

The imputation of missing data related to emission factors is similar to the above-mentioned processes, while the dynamic natures of fossil coal (e.g., sulfur content and ash content) are further taken into account by using provincial trends of variation. The imputation of missing pollutants control rate is more complicated, which involves a dynamic model considering emission standards and retrofitting progress. For units with available end-of-pipe control information meeting current emission standards, we fill the blank spots by using prior information. In cases of other missing values, the emission standard is used to calculate the pollutants control rate, with priority given to installing end-of-pipe control devices for young units with large installed capacity. The model undergoes iterative processes of self-correction until it aligns with the principle that the imputed end-of-pipe control information (i.e., retrofitting progress and average control level) corresponds with the statistical data.”

Supplementary Table 3. Ratio of Blank spots in CPED

Data type	Parameter	Ratio of blank spots
Basic information	Installed capacity	0.2%
	Operating time	0.5%

Operating conditions	Capacity factor	2.2%
	Generation	2.1%
	Coal consumption	1.7%
Emission information	Emission factor	3.4%
	Pollutants control rate	4.7%

Specific comment 9:

Some work should be done to revise the writing style of the paper. At the moment it is a bit too prescriptive and has some stylistic issues. It will benefit from grammar and style correction from a native speaker.

Response: We thank the Referee for the valuable comment. We have spent much time examining, editing, and rewriting the main text. We believe the linguistic quality and clarification have been improved in the revision.

Specific comment 10:

Other articles of interest for you might be

“Radical transformation pathway towards sustainable electricity via evolutionary steps.” Nature communications 10.1 (2019): 1-16.

“Quantifying stranded assets of the coal-fired power in China under the Paris Agreement target.” Climate Policy 23.1 (2023): 11-24.

“Estimating stranded coal assets in China’s power sector.” Utilities Policy 75 (2022): 101352.

“Coupling effects of cross-region power transmission and disruptive technologies on emission reduction in China.” Resources, Conservation and Recycling 189 (2023): 106773.

Response: We thank the Referee for pointing out these interesting articles. We have enriched the main text according to the data and conclusions provided by these papers, and we have also added them as references.

Reviewer #2 (Remarks to the Author):

The authors explored the cost-effectiveness uncertainty brought by policy implementation disturbance of different phaseout and new-built strategies (i.e., the disruption of phaseout priority) of coal power based on a developed unit-level uncertainty assessment framework, and revealed the opportunity and risk of coal transition decision by employing preference analysis. Overall, the topic is interesting, timely and falls within the scope of this journal. I would like to take a further review and probably cite this article in my future work if the following concerns are well addressed.

Response: We thank the Referee for the positive view on the topic and thoughtful suggestions below. We've made a number of revisions to substantially improve the work and address the concerns. A point-to-point response is presented below.

1. Considering the demand of coal power also influences how coal power transition. How do the authors consider the impact of this effect on the result?

Response: We thank the Referee for the valuable comment. We totally agree the demand of coal power influences how coal power transition. A sensitivity test of coal power demand has been conducted as suggested. Our assumption of coal power demand trend for this study is based on the positive projection of CCUS development aiming to carbon neutrality by 2060 (Cheng et al., 2023), therefore, which is relatively higher than some studies that assumes faster paces of renewable development (Zhang et al., 2022; Zhuo et al., 2022; Yu et al., 2022).

Here, we selected another low coal power demand pathway harmonized from Zhang et al. (2023) to further assess the impact. The result indicates that despite a significant decrease in CO₂ emissions and health risks under lower power demand scenario for coal power, the disparity between strategies remains unchanged (Supplementary Fig. 18b-c). The overall difference in net benefits among strategies is not significant in another mitigation trajectory of coal power demand (Supplementary Fig. 18d). When it comes to preference-based phaseout decision, the alterations in power demand have resulted in minimal changes compared to the original scenario (Supplementary Fig. 18). These minimal changes can be attributed to the limited disruption of power demand on the gap of benefits and costs among strategies.

A description about "Sensitivity test of future coal power demand" has been added in the revised Discussion (Line 285-Line 288) and Supplementary Text 7 as suggested, as shown below.

Discussion

"Our test of a lower coal power demand (82% of original demand in 2030) indicates that the relative gaps of benefits and costs among strategies has not changed significantly as the same assumption for phaseout and new-built mechanism (Supplementary Text. 7; Supplementary Figs. 18 and 19)."

Supplementary Text 7. Sensitivity test of future coal power demand

“Existing research offers diverse mitigation pathways for coal-fired power plants in China, some of which suggest generation of coal power will decrease to lower levels, driven by accelerated renewable development⁸⁻¹⁰. A sensitivity test is performed here to assess the impact on benefits trend and phaseout decision under a more ambitious mitigation trajectory of coal power generation, by incorporating and harmonizing another generation projection trend of Zhang et al., 2023¹¹ from 2021-2060 (Supplementary Fig. 18a).

The disparity between strategies remains unchanged, despite a significant decrease in CO₂ emissions and health risk related to coal plants (Supplementary Fig. 18b-c). Taking Death strategy for example, median CO₂ emissions could decrease from 80,200 (CI, 78,600-81,800) to 62,600 (CI, 61,100-64,300) in 2030 in a lower power demand. Meanwhile, the CO₂ emission ratio between Carbon and Death strategy keeps stable in 0.98 within the original scenario and lower power demand. Similarly, the net benefit trends benchmarked against Historical strategy exhibit minor changes (Supplementary Fig. 18d). For example, the changing ratio of median net benefit of Death strategy is only 6% in both scenarios. This is because the overall characteristics of coal power fleet (including existing and new-built units) remain unchanged, even with a slight variation in generation scenarios. This preserves the differences in benefits within each strategy and leads to minimally influence on the result of preference analysis (Supplementary Fig. 19).

”

Supplementary Figure 18 | Trends of coal power generation, CO₂ emissions, deaths, and benefits for each strategy under difference mitigation trajectory. (a) Projection of coal power generation in China (b) CO₂ emissions. (c) Coal-power-related premature deaths. (d) Monetized benefits compared to the historical phaseout pathway. The solid lines in b-d represent the median value of each indicator under each strategy.

Supplementary Figure 19 | Preference analysis for coal power phaseout strategy design.

Coal power phaseout strategy selection according to the maximum normalized total benefits original scenario (a) and sensitivity test of lower power demand (b). α , β , γ represents the preference weighting factor of climate change mitigation, public health protection and assets stranding avoiding, respectively. The higher value of the specific preference weighting factor, the more emphasis is placed on the that preference.

References:

Zhang, S. & Chen, W. Y. China's Energy Transition Pathway in a Carbon Neutral Vision. *Engineering-Prc* **14**, 64-76 (2022). <https://doi.org:10.1016/j.eng.2021.09.004>

Zhuo, Z. Y. *et al.* Cost increase in the electricity supply to achieve carbon neutrality in China. *Nature Communications* **13** (2022). <https://doi.org:ARTN 317210.1038/s41467-022-30747-0>

Yu, S. *et al.* Synthesis Report 2022 on China's Carbon Neutrality: Electrification in China's Carbon Neutrality Pathways. (2022).

Zhang, K., Zhang, W., Shi, Q., Zhang, J. & Yuan, J. Coupling effects of cross-region power transmission and disruptive technologies on emission reduction in China. *Resources, Conservation and Recycling* **189**, 106773 (2023). <https://doi.org:https://doi.org/10.1016/j.resconrec.2022.106773>

2. Whether then development of clean energy needs to be taken into account in evaluating the decision to phaseout and new construction.

Response: We thank the Referee for the valuable and thoughtful comment. Clean energy has been considered in the climate scenario of Carbon Neutrality develop by Cheng *et al.*, 2023, where the penetration of non-fossil-fired power generation would reach 43% in 2030 and 87% in 2060.

A lower coal power demand, meaning the higher penetration of clean energy, was incorporated into our modelling framework to simulate how advancement in technology or the expansion of renewable energy would drive the turnover of future coal power fleet (Supplementary Text 7).

Sensitivity test for lower coal power demand has already been covered in the response of comment 1. The result shows that the overall difference in net benefits among strategies and the results of preference-based decision is not significant in another mitigation trajectory of coal power demand, for its limited disruption of power demand on the gap of benefits and costs from phaseout among strategies.

A description about “Consideration of development of clean energy” has been added in the revised Discussion section (Line 282-Line 288) as suggested.

Discussion

“First, various coal power generation scenarios (i.e., reflecting the degrees of clean energy development) may introduce uncertainties due to the insufficient temporal resolution of integrated assessment models (IAMs) and the overlooked feedback of changing cost effectiveness when implementing different phaseout strategies. Our test of a lower coal power demand (82% of original demand in 2030) indicates that the relative gaps of benefits and costs among strategies has not changed significantly as the same assumption for phaseout and new-built mechanism (Supplementary Text. 7; Supplementary Figs. 18 and 19).”

3. A bit more background and mathematical description on death intensity definitions should be given.

Response: We thank the Referee for the constructive comments. The background and detailed definition of “death intensity” has been supplemented as suggested, which represents the relationship between unit characteristics (installed capacity, MW) and health risk (deaths). The indicator of death intensity (deaths per MW) highlights the notable heterogeneity in health risks among units and emphasizes the co-benefits of health-targeted strategy, particularly when phasing out units with comparable installed capacities.

The concept of death intensity is indeed inspired by the definition of emission intensity (tonnes emissions per MW) used in previous studies to portray the relationship between generating capacity and environmental or climate impacts (Tong et al., 2018 and 2021; Xu et al., 2023).

For the estimate of death intensity, we first obtained the installed capacity in MW directly from the CPED database, and then used the Geos-Chem Adjoint model and Global Exposure Mortality Model (GEMM) to separate the PM_{2.5}-related premature mortality attributed to each generating unit. Subsequently, death intensity is defined by dividing unit-level premature mortality by installed capacity (see equation R1 below). A more detailed background and mathematical description about death intensity definition has been added in the revised Result (Line 97-Line 101) and Method sections (Line 368-Line 376 and Line 429-Line 434).

Unit-level heterogeneity

“Across all categories of capacity, CO₂ emission intensity (defined as CO₂ emissions per capacity, highlighting the heterogeneity in CO₂ emissions) and the death intensity (defined as deaths per capacity, highlighting the heterogeneity in health risks; see Methods in detail) of

small units (i.e., ≤ 100 MW) are 1.4 and 4.0 times larger than the national average level (Supplementary Fig. 2).”

Mortality estimation.

“The assessment of premature mortality attributable to $PM_{2.5}$ exposure necessitates an understanding of the relationship between chronic exposure and response (C-R). Burnett et al.³¹ developed integrated concentration-response functions (IER) for the Global Burden of Diseases Study (GBD), based on cohort studies of ambient air pollution, first- and second-hand tobacco smoking, and household indoor air pollution. Although IER are widely applied to calculate $PM_{2.5}$ -related mortality in research community^{32,33}, non-ambient $PM_{2.5}$ -mortality associations in IER functions and additional causes of death other than the five (i.e., ischemic heart disease, stroke, chronic obstructive pulmonary disease, lung cancer, and lower respiratory infections) considered by the GBD may bias mortality estimation, especially in highly polluting region like China.”

Unit-level health burden decomposition.

“In pervious study, emission intensity (tonnes per MW) is used to depict the relationship between generating capacity and environment impact from each coal-fired units and further identify the “super-polluting” units^{17,41}. Inspired by definition of emission intensity, we defined death intensity (deaths per MW) as unit-level $PM_{2.5}$ -related premature deaths (M_i) per installed capacity (C_i) to highlight the notable heterogeneity in health risks among units⁷, following equation (R1):

$$\text{Death intensity}_i = \frac{M_i}{C_i} \quad (\text{R1})$$

Where C_i represents the installed capacity of units i .”

References:

Tong, D. et al. Targeted emission reductions from global super-polluting power plant units. *Nature Sustainability* **1**, 59-68 (2018). <https://doi.org:10.1038/s41893-017-0003-y>

Tong, D. et al. Health co-benefits of climate change mitigation depend on strategic power plant retirements and pollution controls. *Nature Climate Change* **11**, 1077-1083 (2021). <https://doi.org:10.1038/s41558-021-01216-1>

Xu, R. C. et al. Plant-by-plant decarbonization strategies for the global steel industry. *Nature Climate Change* (2023). <https://doi.org:10.1038/s41558-023-01808>

4. The authors could provide more detail on the amount of data.

Response: We thank the Referee for constructive suggestion. More detailed on the amount of baseline emission data has been supplemented as suggested. We also spent much time rewriting the main text by adding more information and data of the projected phaseout pathways. We believe the linguistic quality and clarification have been improved in the revision.

As the baseline emission dataset in our study, China coal-fired Power plant Emissions Database (CPED) offers detailed year-by-year information on activity data, operating status, emission factors, geographical location, and end-of-pipe control technology. In 2018, CPED covers 5,553 coal-fired generating units, accounting for over 94% of the total installed capacity of coal power. According to CPED, coal power in China played a significant role in greenhouse gas and air pollutant emissions (17.5% of SO₂, 15.6% of NO_x, 3.9% of PM_{2.5}, 34.7% of CO₂ in 2018, detailed in Supplementary Table 5). In addition to existing units, CPED also provides information on retired power units totaling 127 GW, aiding in the characterization of phaseout strategies since the 11th Five-Year Plan (Extended Data Figure 2).

More detailed on the background and the data amount of baseline emission inventory are added in revised Method section (Line 360-Line 367) and Supplementary Text 9.

Unit-level emission database.

“We employ unit-based information (i.e., locations, unit capacity, start year of operation, coal consumption rate, and CO₂ and pollutants emission) from CPED. Constructed based on unit-level information sourced from the Ministry of Ecology and Environment (MEE, unpublished data), CPED is a high-resolution emission inventory that provides year-by-year emissions from coal-fired power plants since 1990 in China developed by Tsinghua University. More details about CPED can be found in Supplementary Text 8 and our previous studies by Liu et al.⁴ and Tong et al.³⁰. A detailed description about the preprocessing of baseline emission inventory has been added in the Supplementary Text 9.”

Supplementary Text 8. China coal-fired Power plant Emissions Database

“China coal-fired Power plant Emissions Database (CPED) is developed and maintained by Tsinghua University and keeps tracking emissions of coal-fired power plants for a 30-year period. Built upon data from Ministry of Ecology and Environment (MEE; unpublished data), CPED offered year-by-year detailed information about activity data, operating status, emission factors, geographical location, end-of-pipe control technology. More detailed about CPED can be found in Liu et al., 2015¹² and Tong et al., 2018¹³. Previous studies have verified that the magnitude and trends of power emissions in CPED are in good agreement with top-down estimates from satellite measurements.

CPED used here covers 5,553 coal-fired generating units with 947.8 GW (94% of the total installed capacity) in 2018. According to CPED, coal power consumed more than 50% of the total production, accounts for more than 60% of total national power generation, and contributes significantly to the total emission of CO₂ and air pollutants (17.5% of SO₂, 15.6 of NO_x, 3.9% of PM_{2.5}, 34.7% of CO₂ in 2018, see Supplementary Table 5). Coal-fired power plants are primarily located in provinces with abundant resources, high demand, developed industries, and convenient transportation (e.g., Jiangsu, Inner Mongolia, Shandong, Shanxi). This distribution overlaps with densely populated areas in China, intensifying the health risks associated with pollutant emissions.

Additionally, CPED also provides information on 127 GW retired coal power capacity, which helps to characterize the prior phaseout practice since the 11th Five-Year-Plan (Extended Data Figure 2).”

Extended Data Figure 2 | Accumulative capacity by individual retired units for coal power, ranked by the phaseout year and ratio of Age-to-Capacity or coal consumption rate. (a-b) The bar represents the cumulative capacity of individual generating units in order, ranking by phaseout year and ratio of age to capacity. (a) bars are distinguished by capacity group, (b) bars are distinguished by age group. (c) is similar to (a) and (b), but units are ranked by both phaseout year and coal consumption rate, and distinguished by coal consumption rate group.

Supplementary Table 1 | Emission and key parameters of China’s coal power plants fleet in 2018.

Category	Subcategory	2018
Activity data	Coal consumption (Mt)	2022.2

	Power generation (TWh)	4312.6
	Coal consumption rate (gce/kWh)	307.6
Capacity sizes	<100 MW	6.5%
	100~300 MW	9.6%
	300~600 MW	36.7%
	≥600 MW	47.1%
Average removal efficiency	De-SO ₂ device	94.0%
	De-NO _x device	73.0%
	De-PM _{2.5} device	98.9%
Emissions	SO ₂ (Tg/yr)	1.65
	PM _{2.5} (Tg/yr)	0.26
	NO _x (Tg/yr)	3.33
	CO ₂ (Tg/yr)	3525.4

5. Which policy scenario is most appropriate for coal power transition in China, and how feasible is the policy in this scenario?

Response: We thank the Referee for this thoughtful question. The BAU strategy is developed based on the historical patterns of coal power transition, revealing that small, old, or inefficient generating units are prioritized for phaseout (see Extended Data Fig. 2). This strategy might continue to act as the most feasible scenario because it could effectively balance emission mitigation and economic considerations, according to prior practice and scenario analysis.

Since China has spared no effort in controls of air pollutions for coal power plants for over 10 years, narrowed disparities of emissions and new attention of climate and health impacts pose challenges to the refined governance of coal power. BAU strategy might not be as effective, considering the potential future shifts in decision preferences. Thus, our study endeavors to objectively empathize the potential impact of cost effectiveness uncertainty in phaseout decision and analyze the strengths and weaknesses of each strategy in addressing different decision preferences.

Phaseout decision is still a preference-based open issue, especially when it comes to local level. Taking Inner Mongolia for example, Age-to-Capacity strategy might be the most suitable for its young power fleet (Supplementary Fig. 17). A detailed description about “appropriate strategy for coal power transition” and “tailoring preference-based phaseout decision at a finer level” has been added in the revised Discussion section (Line 273-Line 280).

For further research, we hope to acquire more data and incorporate additional factors into our data-driven dynamic uncertainty assessment framework to enhance the adaptivity and feasibility of coal power transition in the future. This aims to provide policy-makers with a more comprehensive and feasible understanding of the risk and potential outcomes of their preference-based decisions.

Discussion

“In alignment with the historical patterns of phaseout decision, BAU might remain the most appropriate and feasible phaseout strategy in future coal power transition, for its advantages in balancing emission mitigation and economy. However, considering the changing preferences and above-mentioned time-varying external factors, phaseout decision is still an open issue. Especially when it dives to the local level, decision makers should tailor phaseout plans according to the specific characteristics of coal power fleet. For example, Age-to-Capacity strategy might be more suitable for Inner Mongolia than BAU strategy as its young and huge coal fleet (Supplementary Text 6; Supplementary Fig. 17).”

Reviewer #3 (Remarks to the Author):

The present paper dwells on the intricacies of transition from fossil fuel-based electricity generation to renewables in China. Recent evidences cited in the paper suggest that implementation of the 11th Five Year Plan led to closure of 127 GW of relatively inefficient coal-based Electricity Generating Units (EGUs), while at the same time more efficient EGUs are also slated to start production in the future. In this context, the work tries to derive compatible pathways to phaseout coal based EGUs.

Response: We thank the Referee for the positive and accurate summary of our work.

There can be multiple criteria to determine the phaseout strategies. The authors adequately address them by considering five phaseout strategies: Historical strategy (allow plants to operate for their historical expected lifetime); BAU strategy (consistent with existing phase out strategy); Death strategy (prioritize public health protection); Carbon strategy (prioritize climate change mitigation) and Age-to-Capacity strategy (prioritize economic loss avoidance). After providing weights to different strategies, the authors use multiple criteria decision making (MCDM) method in an attempt to generate pareto-optimal solutions of strategic options suitable for China. Use of this methodology is appropriate to deal with the uncertainties in the model. The estimates with their associated confidence intervals are also reported in an adequate manner.

Response: We thank the Referee for the positive and accurate summary of our work and for the fair and thoughtful comments below. We have added a regional case study, made sensitivity tests for health co-benefits of coal power transition in response, and believe the manuscript has been substantially improved. A point-to-point response is presented below.

Notably, the authors use α as the weight for climate change mitigation strategy, β as the weight for health benefits and γ as weight for stranded asset avoidance. The authors suggest to use a “dimensionless indicator (named the normalized net benefit) obtained by normalizing and weighed summing the above three factors” to help the decision makers in strategy assessment based on their time-varying preferences. Obviously, the idea holds merit. May the authors be asked to demonstrate the workability of the idea by calculating the same on a limited dataset of a particular region of China?

Response: We thank the Referee for the positive tone and constructive comment. A particular region of Inner Mongolia in China has been selected as suggested under the comprehensive considerations of two factors. First, Inner Mongolia owns relatively youth and huge coal power fleet compared to other provinces, with total capacity of 79.8 GW (rank second in 31 provinces) and averaged age of 9.3 in 2018 (~10.4 years on average in 31 provinces). Second, the abundant coal resources make Inner Mongolia a potential future energy base, which in turn emphasize the need for cautious planning in the phaseout and new-built decision of coal-fired power plants.

The result of case study in Inner Mongolia shows that the young fleet of coal power would amplify the risk of negative outcomes for most strategies because of huge asset stranding. Strategies targeted health protection or asset stranding avoidance are likely to ensure net positive benefit in future turnover. Different from the preference-based decision making in

national level, BAU strategy might not be suitable for most decision preferences in Inner Mongolia for its moderate performance in environmental protection and related areas.

Case study in Inner Mongolia not only highlights the importance of preference-based decision making, but also underscores the crucial role of tailoring phaseout plans to the specific characteristics of coal power fleet at a finer scale. A detailed description about “Case study of Inner Mongolia” has been added in the revised Result (Line 242-Line 244), Discussion section (Line 276-Line 280), and Supplementary Text 17 as suggested.

The bias of strategy selection based on decision preferences

“And uncertainty would still have an undeniable impact on the preference-based decision making at the local scale (the case study in Inner Mongolia; Supplementary Text 6; Supplementary Fig. 15).”

Discussion

“Especially when it dives to the local level, decision makers should tailor phaseout plans according to the specific characteristics of coal power fleet. For example, Age-to-Capacity strategy might be more suitable for Inner Mongolia than BAU strategy as its young and huge coal fleet (Supplementary Text 6; Supplementary Fig. 17). ”

Supplementary Text 6. Case study of Inner Mongolia

“Coal power will be a crucial lever for Inner Mongolia to achieve clean air and climate mitigation. Benefiting from abundant coal and renewable resources, as well as developed external power transmission grids, Inner Mongolia consistently takes the lead in outbound electricity since 2013 and becomes a major energy base in northern China. With a relatively young and extensive coal power infrastructure (total capacity of 79.8 MW and averaged age of 9.3 in 2018), coal power generation in Inner Mongolia reached 388 TWh in 2018, ranking third nationally after Shandong and Jiangsu. The low-carbon and green transition of coal power holds significant importance in this region, due to the coal-dominated energy structure and high proportion of industrial electricity usage. Therefore, we choose Inner Mongolia as a case to assess the potential outcomes and its uncertainty of coal power phaseout policy.

Inner Mongolia might face a unignorable risk of achieving negative net benefits (Supplementary Fig. 15a). Taking BAU strategy as an example, the probability of achieving negative net benefits is 30%, much higher than the national average level. This is due to the huge asset stranding when implementing early phaseout of young power fleet in Inner Mongolia. Nevertheless, Death and Age-to-Capacity strategies are likely to ensure net positive net benefit in future turnover. Such targeted phaseout strategies can accelerate the replacement of small power plants with cleaner and larger ones, leading to an increasingly clustered distribution of coal power (Supplementary Fig. 15b). Despite the increasing coal power demand, the number of coal-fired power units in Age-to-Capacity have decreased from 378 to 284 during 2018-2030, with an average installed capacity per unit reaching 500 MW. Within Historical strategy, there would still be 207 units with installed capacity below 300 MW in 2030.

Supplementary Figure 17 shows phaseout strategies selection tailored to different policy preference in Inner Mongolia. The prior phaseout practice (i.e., BAU strategy) might not be

the optimal choice given various decision preferences (Supplementary Figure 17a), which is different from the nationwide preference-base phaseout decision. An alternative and suitable phaseout strategy should be sought for the future decarbonization of coal power, according to the objectives set by policy-makers. Uncertainties may still bias the preference-specified phaseout decision to a certain degree in Inner Mongolia (Supplementary Figure 17b-d). Therefore, it is necessary to thoroughly assess the emissions and operational conditions of all units and implement such a unit-by-unit phaseout plan in a finer level.”

Supplementary Figure 15 | Cumulative net benefits and the potential turnover of power fleet for each strategy in Inner Mongolia. (a) Cumulative net benefits of targeted strategies represent the sum of monetized CO₂ emission reduction benefits and health co-benefits minus the costs of asset stranding. (b) The evolution of coal-fired power plants in 2018, 2030, and 2060 for each strategy, respectively.

Supplementary Figure 17 | Preference analysis for coal power phaseout strategy design tailored to Inner Mongolia. (a) Coal power phaseout strategy selection according to the maximum normalized total benefits. α , β , γ represents the preference weighting factor of climate change mitigation, public health protection and assets stranding avoiding, respectively. The higher value of the specific preference weighting factor, the more emphasis is placed on the that preference. (b-e) Disturbed phaseout strategy decisions, which are disturbed by the pathway with 95th percentile of monetized net benefits under BAU (b), Death (c), Age-to-Capacity (d), and Carbon (e) strategies, respectively.

In reference to the Death strategy, it may be noted that exposure to PM_{2.5} leading to mortality is calculated in this paper using the GEMM as developed by Burnett et al. The model may be using concentration response functions to estimate the relative risk of deaths due to non-communicable diseases and lower respiratory infections. Recent advances in the literature suggest that exposure to sulphur, sulphates or PM_{2.5} from coal emissions are associated to higher morbidity in comparison with PM_{2.5} from other sources per unit concentration. Recent studies (Henneman et al) shows that exposure to coal PM_{2.5} was associated with 2.1 times greater mortality risk then exposure to PM_{2.5} from all other sources. Maybe, the death estimates by the authors of this paper are biased downward in the light of the above revelation.

Response: We thank the Referee for pointing out this very important aspect. We totally agree that health co-benefits would be underestimated if PM_{2.5} from coal emissions are associated to higher morbidity in comparison with PM_{2.5} from other sources per unit concentration reported by the recent Science study (Henneman et al., 2023). Sensitivity tests of higher morbidity (Supplementary Text 10) have been added according to American exposure relationship due to the absence of a China's localized concentration-response function for the coal power plants.

The sensitivity test shows that the underestimation of health effects does have a noticeable impact on the overall assessment of effects (Supplementary Fig. 20). The overall net benefits of Carbon strategy increased to 448 (CI, 134~758) billion RMB when applying correction factor of 2.1 in the estimation of health risk, while Carbon strategy can hardly get a positive outcomes in the origin scenario. Within the framework of the Healthy China Initiative, it is crucial to promptly update health assessment tools to improve the accuracy of related policy assessment. However, the preference-based phaseout decision shows limited changes compared to the original scenario (Supplementary Fig. 24). This can be attributed to the stability in relative rankings and differences of normalized health co-benefit among various phaseout pathways, even after applying the correction factor. The related discussion on the uncertainty of morbidity for the power plants has been added in the revised Discussion section (Line 295-Line 299) and Supplementary Text 10 as suggested.

Discussion

“Fourth, health risk assessment based on GEMM and GEOS-Chem adjoint model may arise certain uncertainties for their incomplete understanding of mortality risk, chemical and physical processes, future meteorological conditions, and structural evolution in the population, leading to a potential underestimation of the health co-benefit²⁹ (Supplementary Text 10; Supplementary Fig. 20).”

Supplementary Text 10. Sensitivity test of health risk estimation

“Traditional concentration-response functions, which the lack of availability of source-specific exposure estimates, are likely to underestimate mortality burden attributed to coal power⁹. Advanced research based on U.S.A. Medicare beneficiaries reported that exposure to coal PM_{2.5} may lead to a mortality risk approximately 2.1 times greater than exposure to PM_{2.5} from all sources¹⁰. Limit to the absence of relevant epidemiological data and tools, our study is temporarily unable to employ a source specified health risk assessment tailored to coal power in China. The potential impact of underestimating health risks on the net benefits of each strategy is assessed by conducting additional sensitivity tests on the health risk attributed to Coal PM_{2.5}.

The underestimation of health effects does indeed have a noticeable impact on the overall assessment of effects (Supplementary Fig. 20). Using the Carbon strategy as an example, the overall net benefits increase from -378 (CI, -528~-231) to 448 (CI, 134~758) billion RMB with a health risk correction factor of 2.1 times, and there is almost no risk of negative outcomes. The implementation of those targeted strategies may yield unexpected magnitude of overall benefits, compared with the historical one.

Limited changed is observed in the preference-based phaseout decision compared to the original scenario (Supplementary Fig. 24). This is because the correction factor on the health risk of coal PM_{2.5} had no impact on the relative rankings and differences of normalized health co-benefit among different phaseout pathways, which is adheres to the principle that health co-benefit might only play a secondary role as indirect outcomes under policy preferences devaluing health considerations.

In response to the call for Healthy China Initiative, we believe that health protection will become a pivotal consideration in future decision-making. There is an urgent need to update the coal-specified health assessment methodology and reduce associated uncertainties.”

Supplementary Figure 20 | Comparison of the cumulative net benefits under original and 2.1 times health risk.

Supplementary Figure 24 | Preference analysis for coal power phaseout strategy design.

Coal power phaseout strategy selection according to the maximum normalized total benefits under original (a) and 2.1 times health risk (b). α , β , γ represents the preference weighting factor of climate change mitigation, public health protection and assets stranding avoiding, respectively. The higher value of the specific preference weighting factor, the more emphasis is placed on the that preference.

The present study is static in nature. A longitudinal study that considers the dynamic changes due to economic cycles may lead to alteration of ranking of power plants closures under the strategies considered in the paper. Economic downturns may lead to lower demand of electricity and thus lower production from coal plants. Introduction of renewables in the grid may also lead to altered production schedule and lesser usage of these coal based generating units. To address these issues, the authors may be encouraged to run the modelling exercise periodically, more specifically quinquennially in sync with the Five-year Plans of China.

Response: We thank the Referee for constructive suggestion. We totally agree that the coal transition decision would be dynamically affected by socioeconomic changes, technology innovation, and implemented policies such as economic downturns, accelerated renewable development. A regular update in a 5-year step (in line with the Five-year Plans of China) is necessary to address these emerging issues, therefore, as suggested, we will incorporate the latest unit-level information into our dynamic uncertainty assessment framework for coal power transition and keep tracking the cost-effectiveness uncertainty of preference-based coal transition decision.

We have also added discussion on above-mentioned affected factors of coal transition decision and this update plan in the revised main text (Discussion section, Line 307-Line 313) as suggested.

Discussion

“Recognizing the dynamic nature of phaseout decision affected by variable factors such as socioeconomic changes, technology innovation, and implemented policies, we will incorporate the most up-to-date unit level information into our modelling framework and keep tracking the cost-effectiveness uncertainty of preference-based coal transition decision in a 5-year step (in

line with the Five-year Plans of China). This, in turn, enhances the robustness, adaptivity, and applicability of cautiously designed phaseout strategies, ensuring that they remain promising cost-effective in response to different policy preferences.”

The plan of the work is very well laid out and executed in a very comprehensive way.

Response: We thank the Referee for the positive tone of our work.

Reference

Lucas Henneman et al., Mortality risk from United States coal electricity generation. *Science* 382, 941-946 (2023). DOI:10.1126/science.adf4915

(S.Majumdar)

Response: We have cited this work for health benefits comparison and added this reference as suggested.

REVIEWERS' COMMENTS

Reviewer #1 (Remarks to the Author):

The authors have answered my review issues , the article can be accepted!

Reviewer #2 (Remarks to the Author):

I think the authors have well solved my concerns. I have no more suggestion.

Reviewer #3 (Remarks to the Author):

I am satisfied with the article in the current form as all the points raised by me has been adequately addressed. The paper is also now more readable.

Reviewers comments:

Reviewer #1 (Remarks to the Author):

The authors have answered my review issues, the article can be accepted!

Response: We thank the Referee for the positive comments and recommendation about the revised manuscript. And we also thank the Referee for the time and efforts s/he devoted to improving our manuscript.

Reviewer #2 (Remarks to the Author):

I think the authors have well solved my concerns. I have no more suggestion.

Response: We thank the Referee for the positive comments and recommendation about the revised manuscript. And we also thank the Referee for the time and efforts s/he devoted to improving our manuscript.

Reviewer #3 (Remarks to the Author):

I am satisfied with the article in the current form as all the points raised by me has been adequately addressed. The paper is also now more readable.

Response: We thank the Referee for the positive comments and recommendation about the revised manuscript. And we also thank the Referee for the time and efforts s/he devoted to improving our manuscript.